



# Multiphase MCM/CAPRAM modeling of formation and processing of secondary aerosol constituents observed at the Mt. Tai summer campaign 2014

Yanhong Zhu[1,2,a], Andreas Tilgner[2], Erik Hans Hoffmann[2], Hartmut Herrmann[2,3*], Kimitaka Kawamura[4,b], Lingxiao Yang[1], Likun Xue[1*], Wenxing Wang[1]

[1]Environment Research Institute, Shandong University, 266237 Qingdao, China
[2]Leibniz Institute for Tropospheric Research (TROPOS), Atmospheric Chemistry Department (ACD), 04318 Leipzig, Germany
[3]School of Environmental Science and Engineering, Shandong University, 266237 Qingdao, China
[4]Institute of Low Temperature Science, Hokkaido University, Sapporo 060-0819, Japan
[a]Now at: Department of Atmospheric Sciences, School of Earth Sciences, Zhejiang University, 310012 Hangzhou, China
[b]Now at: Chubu Institute for Advanced Studies, Chubu University, Kasugai 487-8501, Japan

*Correspondence to:* Likun Xue (xuelikun@sdu.edu.cn), Hartmut Herrmann (herrmann@tropos.de)

**Abstract.** Despite the high abundance of secondary aerosols in the atmosphere, their formation mechanisms remain poorly understood. In this study, MCM/CAPRAM mechanism is used to investigate the multiphase formation and processing of secondary aerosol constituents during the advection of air masses towards the measurement site of Mt. Tai in North China. Trajectories with and without chemical cloud interaction are modeled. Modeled radical and non-radical concentrations demonstrate that the summit of Mt. Tai, with an altitude of ~1.5 km a.m.s.l., is characterized by a sub-urban oxidants budget. The modeled maximum gas-phase concentrations of OH radical are $3.2 \times 10^6$ molecules cm$^{-3}$ and $3.5 \times 10^6$ molecules cm$^{-3}$ in simulations with and without cloud passages in the air parcel, respectively. Different to previous studies at Mt. Tai, this study has modeled chemical formation processes of secondary aerosol constituents under day vs. night and cloud vs. non-cloud cases along the trajectories to Mt. Tai in detail. The model studies show that sulfate is mainly produced in simulations where the air parcel is influenced by cloud chemistry. Under the simulated conditions, the aqueous reaction of $HSO_3^-$ with $H_2O_2$ is the major contributor to sulfate formation, contributing 67 % and 60 % in the simulations with cloud and non-cloud passages, respectively. The modeled nitrate formation is higher at nighttime than at daytime. The major pathway is aqueous-phase $N_2O_5$ hydrolysis, with a contribution of 72 % when cloud passages are considered and 70 % when not. Secondary organic aerosol (SOA) compounds, e.g. glyoxylic, oxalic, pyruvic and malonic acid, are found to be mostly produced from the aqueous oxidations of hydrated glyoxal, hydrated glyoxylic acid, nitro 2-oxopropanoate and hydrated 3-oxopropanoic acid, respectively. Sensitivity studies reveal that gaseous VOC emissions have a huge impact on the concentrations of modeled secondary aerosol compounds. Increasing the VOC emissions by a factor of two leads to linearly increased concentrations of the corresponding SOA compounds. Studies using the relative incremental reactivity (RIR) method have identified isoprene, 1,3-butadiene and toluene as the key precursors for glyoxylic and oxalic acid, but only isoprene is found





to be a key precursor for pyruvic acid. Additionally, the model investigations demonstrate that an increased aerosol
partitioning of glyoxal can play an important role in the aqueous-phase formation of glyoxylic and oxalic acid. Overall, the
present study is the first that provides more detailed insights in the formation pathways of secondary aerosol constituents at
Mt. Tai and clearly emphasizes the importance of aqueous-phase chemical processes on the production of multifunctional
carboxylic acids.

## 1    Introduction

Secondary aerosols are more abundant than primary aerosols (Volkamer et al., 2006). Their constituents are formed on a
regional scale and transported over long distances and thus have a direct impact on the air quality of a wider area (Kim et al.,
2007; Matsui et al., 2009; DeCarlo et al., 2010). Secondary aerosols are usually divided into two classes: secondary
inorganic aerosol (SIA) and secondary organic aerosol (SOA). A number of studies have been conducted aiming at
investigating their formation mechanisms (Yao et al., 2002; Duan et al., 2006; Wang et al., 2006; Guo et al., 2010; Zhao et
al., 2013). The SIA components, including sulfate, nitrate and ammonium, are important contributors to fine particulate
matter (PM$_{2.5}$) and play an important role in haze formation (Volkamer et al., 2006; Sun et al., 2014; Wang et al., 2014;
Zhang et al., 2014). The SIA formation processes are relatively well understood, but some indefiniteness still remains, such
as multiphase formation, particularly under highly polluted conditions such as in China (Wang et al., 2014; Wang et al.,
2016a). SOA is also a key component of PM$_{2.5}$ and linked to adverse health effects, visibility reduction and climate change
(Tabazadeh, 2005; Seagrave et al., 2006; De Gouw and Jimenez, 2009). However, their formation mechanisms are still not
well understood (Huang et al., 2014).

Dicarboxylic acids and related compounds (oxo-carboxylic acids and α-dicarbonyls) (DCRCs) are ubiquitous water-soluble
components of SOA (Kawamura and Sakaguchi, 1999; Kawamura and Yasui, 2005; Pavuluri et al., 2010). They are mainly
produced by secondary processes of precursors via gas-phase and subsequent aqueous-phase reactions (Glasius et al., 2000;
Legrand et al., 2007; Kundu et al., 2010; Tilgner and Herrmann, 2010). A detailed knowledge of the formation processes of
DCRCs is helpful to better understand the fate of SOAs in the troposphere. A number of studies have proposed that aromatic
hydrocarbons, isoprene and ethene are important precursors for DCRCs (Warneck, 2003; Ervens et al., 2004; Bikkina et al.,
2014; Tilgner and Herrmann, 2010). However, formation pathways based on measured data are still limited. Additionally,
model studies show growing evidence that substantial amounts of DCRCs are formed by aqueous-phase reactions within
aerosol particles, clouds and fog droplets (Sorooshian et al., 2006; Carlton et al., 2007, 2009; Ervens et al., 2008, 2011;
Ervens, 2015, Tilgner and Herrmann, 2010, Tilgner et al., 2013). Nevertheless, the applied mechanisms are still incomplete
and the formation processes are therefore not completely understood. Hence, in this study, a near-explicit multiphase model
is applied to investigate the chemical processing of DCRCs in both gas and aqueous phases in order to understand the
formation processes and the fate of DCRCs in the atmosphere.



The present study focuses on the multiphase formation mechanism of key secondary aerosol constituents measured in June 2014 at Mt. Tai, which is the highest mountain in the North China Plain (NCP). Mt. Tai is located in the Shandong province in the NCP, between the Yangtze River Delta and the Bohai Rim, two of China's three largest economic zones with a population of more than 410 million. In summer, clouds frequently occur over the summit of Mt. Tai. Despite a little emission from temples and small restaurants at Mt. Tai's peak, the sampling site on top of Air Force Hotel, Houshiwu was

typically not much influenced by tourists and temples (Sun et al., 2016). The special altitude and geographical location of Mt. Tai provide a suitable site to measure regional secondary aerosol constituents and to investigate their formations pathways along the advection to the measurement site.

The detailed objectives of the present study are as follows: (i) characterization of modeled radical and non-radical oxidant concentrations; (ii) assessment of modeled concentrations and formation processes of key secondary inorganic compounds;

(iii) study of modeled concentrations of DCRCs and a comparison with field observations to assess the model predictions; (iv) investigation of source and sink pathways of selected DCRCs; (v) examination of the impact of emission data on modeled secondary aerosol concentrations; (vi) identification of the key precursors of selected DCRCs and (vii) the impact of higher glyoxal (Gly) partitioning constants on the modeled concentrations of Gly, glyoxylic acid ($\omega C_2$) and oxalic acid ($C_2$).

## 2     Multiphase modeling and model setup

Detailed descriptions about the sampling site, the sampling instruments and the analysis methods can be found in a previous publication (Zhu et al., 2018). Campaign observation data, meteorological conditions and corresponding findings are also given there. The meteorological data during the campaign were as follows: temperatures ranged from 10 to 25 °C with an average of 17 °C; relative humidity (RH) ranged from 58 to 100 % with an average of 87 %; the prevailing wind direction

was northwest; wind speeds ranged from 1 to 7 m/s. The weather conditions were mostly cloudy and occasionally foggy. DCRCs exhibited mostly similar concentrations at daytime and at nighttime, e.g. $C_2$, pyruvic acid (Pyr) and $\omega C_2$ (Zhu et al. 2018). These results differed from previous studies at urban (New Delhi, India) (Miyazaki et al., 2009), suburban (the northern vicinity of Beijing) (He et al., 2013) and mountain sites (Central Himalayan) (Hegde and Kawamura, 2012), where diurnal variations were observed. Therefore, this study focused on the formation processes of these compounds to better

understand the similar concentrations during daytime and nighttime summer conditions at Mt. Tai.

### 2.1     Model and mechanism description

In this study, we applied the air parcel model SPACCIM (SPectral Aerosol Cloud Chemistry Interaction Model, Wolke et al., 2005) to simulate multiphase chemistry along main trajectories during a simulated campaign period. SPACCIM combines a multiphase chemical model with a microphysical model, simulating aqueous-phase chemistry in deliquesced particles and

cloud droplets. More detailed descriptions of SPACCIM can be found in Wolke et al. (2005), Sehili et al. (2005) and Tilgner



et al. (2013). However, SPACCIM cannot assess the complexity of the tropospheric mixing processes along the transport and the effects of non-ideal solutions on the occurring multiphase chemistry. These limitations have to be kept in mind when comparing predicted and observed concentration at Mt. Tai.

The applied multiphase chemistry mechanism is comprised of the Master Chemical Mechanism (MCM3.2 scheme with
13927 reactions, http://mcm.leeds.ac.uk/MCM/, Jenkin et al., 2003; Saunders et al., 2003) and the chemical aqueous-phase radical mechanism (CAPRAM4.0 scheme with 7129 reactions, Bräuer et al., 2019). The MCM3.2 is a near-explicit gas-phase chemistry mechanism, while CAPRAM4.0 explicitly describes the aqueous-phase chemical and phase transfer processes of inorganic compounds and organic compounds of up to 4 carbon atoms. Phase exchange processes (in total 275) are implemented based on the resistance model of Schwartz (1986), in which the mass accommodation coefficient, the gas
phase diffusion coefficient and the Henry's law constant are considered.

## 2.2 Trajectories and performed model simulations

Zhu et al. (2018) have shown that during the sampling period (4 June – 4 July 2014), air masses arriving at Mt. Tai mainly came from the north (named cluster 2) and the south (named cluster 4). The two clusters accounted for 79 % of the total trajectories. Moreover, the sum of DCRC concentrations in clusters 2 and 4 amounted to 73 % of total DCRC concentration
during the sampling period. Therefore, in this study, we selected clusters 2 and 4 to simulate and investigate the formation processes and the fate of DCRCs. Additionally, Zhu et al. (2018) have clearly shown that biomass burning was only important source during the first half of the sampling period (4 – 19 June). In order to minimize the impact of biomass burning including primary emissions, the second half of the sampling period (20 June – 4 July) has been chosen as the model simulation period. In addition, both clusters 2 and 4 exhibited a rather stable transport above the mixing layer to Mt. Tai site.
A total simulation time of 96 h is chosen (4 days), representing a typical aerosol lifespan (Willams et al., 2002). The first 24 h are considered as a model initialization day. Thus, only the model results from 24 h to 96 h are presented in this study. With the help of measured RH at Mt. Tai and meteorological values of clusters 2 and 4 that are obtained by HYSPLIT4.9 (Draxler and Rolph, 2003) and MODIS satellite pictures (Li et al., 2005), we have obtained that clouds most likely occurred at the Mt. Tai top and advected to Mt. Tai at the altitude of the trajectories (Zhu et al., 2018). Radiosonde data
(http://weather.uwyo.edu/upperair/sounding.html) also shows that clouds could occur in the trajectories of clusters 2 and 4. Cloud interactions are considered on the basis of the measured RH values at Mt. Tai and RH values in radiosonde data at about 1500 m. The fractions of RH values above 90 % are calculated and used as a representative for the time possibly spent by a trajectory inside the clouds. For cluster 2, three 1.28 h long daytime and three 1.92 h long nighttime duration cloud cycles are repeated every 24 h. For cluster 4, daytime and nighttime cloud durations are 1.28 h and 1.68 h long, respectively,
and are also repeated every 24 h. In order to better identify the impact of cloud droplet chemistry, we have also considered a model scenario without clouds. During the non-cloud period, RH is 70 % from 24 h to 96 h. Acronyms for the different model simulations performed in this study can be found in Table 1.



We have also carried out sensitivity runs in this study, investigating the following three aspects: (i) the impact of considered emission data on modeled secondary aerosol concentrations; (ii) the identification of key precursors of $C_2$, Pyr and $\omega C_2$ by relative incremental reactivity (RIR) (Xue et al., 2014), and (iii) the assessment of impact of increased Gly aerosol partitioning on concentrations of Gly, $\omega C_2$ and $C_2$. RIR is defined as the ratio of the decrease in the maximum concentrations of the DCRCs and the decrease in the emission data of the corresponding precursor (a 20 % reduction is adopted in this study).

### 2.3 Chemical and physical model initialization

Zhu et al. (2018) have reported that the pollutant concentrations during the campaign at Mt. Tai were largely controlled by long-range transport. The formation processes of secondary aerosols during long-range transport strongly depend on the emission of precursors. Therefore, emission data in clusters 2 and 4 passed over areas are implemented in the model. Biogenic emission data (isoprene, α- and β-pinenes) are obtained from the emissions of the atmospheric compound compilation of ancillary data (http://eccad.sedoo.fr/eccad_extract_interface/JSF/page_login.jsf), while other input emission data (volatile organic compounds (VOCs including alkanes, alkenes, aromatics, aldehydes, alcohols), CO, $CO_2$, $NH_3$, NO, $NO_2$, $SO_2$) are obtained from a new anthropogenic emission inventory in Asia (http://www.meicmodel.org/dataset-mix.html) (Li et al., 2017b). Emission data applied in the model can be found in Table S1. The deposition velocities used in SPACCIM are presented in Table S2. Additionally, the multiphase chemistry mechanism is also constrained by the initial concentrations of key species at the corresponding originated areas, which are obtained from related literature (Table S3 and Table S4). In case of missing values, they are obtained from the CAPRAM urban scenario (http://projects.tropos.de/capram, Ervens et al., 2003). The initial chemical data include gas-phase concentrations of inorganic gases (NO, $NO_2$, $O_3$, $SO_2$, $HNO_3$, $NH_3$, $H_2O_2$), VOCs (including alkanes, alkenes, aromatics, aldehydes, alcohols and ketones) and particle phase data. These initial model data, and also meteorological and aerosol parameters can be found in Table S3 and Table S4.

### 3 Model results and discussion

In this section, first, the modeled concentrations of key radical and non-radical oxidants are discussed. Second, modeled concentrations and formation processes of key secondary inorganic compounds are outlined. Third, DCRC concentrations are discussed and compared with the measured data at Mt. Tai. Then, formation and degradation pathways of selected DCRCs are investigated through detailed reaction flux analyses. Finally, the impact of precursor emissions on the modeled concentrations of secondary aerosol is investigated, and the most important precursors of the selected DCRCs are identified through sensitivity runs. Moreover, the impact of increased aerosol partitioning of Gly on the concentrations of Gly and its oxidation products ($\omega C_2$ and $C_2$) is also examined.





### 3.1 Modeled concentrations of important oxidants

Due to the key role of radical and non-radical oxidations in the formation processes of secondary aerosol constituents, their concentration variations and corresponding reasons are investigated. Several publications have already focused on the
oxidant budget in China. Kanaya et al. (2009) modeled gas-phase concentrations of OH, HO$_2$ and O$_3$ oxidants at Mt. Tai by Regional Atmospheric Chemistry Mechanism (RACM), but they didn't consider the effect of aqueous-phase conditions, such as cloud cases. Kanaya et al. (2013) just observed O$_3$ gas-phase concentration at Mt. Tai, and modeled photochemical O$_3$ production processes by RACM. Suhail et al. (2019) only observed gas-phase NO$_3$ concentration at Wangdu site in the NCP. Ren et al. (2009) and Wang et al. (2016b) reported measurement of gas-phase peroxides at Mt. Tai and Wangdu site in
the NCP, respectively, but no other radical or non-radical oxidants. Liu et al. (2012) modeled gas-phase concentrations of OH, HO$_2$ and RO$_2$, and investigated RO$_x$ budgets and O$_3$ formation in Beijing in the NCP using a 1-D photochemical model. These studies partly investigated the radical or non-radical oxidants, and were lack of aqueous-phase reactions. To our knowledge, this is the first detailed multiphase chemical modeling study examining radical and non-radical concentrations along the trajectory to the Mt. Tai under day vs. night and cloud vs. non-cloud cases.

### 3.1.1 Key radical oxidants

Figure 1 shows the modeled gas- and aqueous-phase concentrations of important radical oxidants in the C2w and C2wo cases. The gas- and aqueous-phase OH, HO$_2$ and NO$_3$ radical concentrations and the corresponding time profiles in the C4w and C4wo cases are quite similar to those in the C2w and C2wo cases. Therefore, the corresponding plots of the C4w and C4wo cases are presented in Fig. S1.

Due to photochemistry, the gas-phase OH and HO$_2$ oxidants showed a distinct diurnal profile (Crosley, 1995), but they are significantly influenced by cloud occurrences, which are consistent with former model studies (Tilgner et al., 2013; Harris et al., 2014; Whalley et al., 2015). Under daytime cloud droplet conditions, OH and HO$_2$ concentrations in the gas phase decreased by about 90 % and 98 %, respectively. The reduction of OH radical is mainly caused by the reduction of the gas-phase formation pathway of the HO$_2$ + NO reaction. Due to high water solubility, a direct phase transfer of HO$_2$ into cloud
droplets contributed significantly to its gas-phase concentration reduction.

The NO$_3$ radical exhibited higher gas-phase concentrations during the night than during the day. The levels are also substantially lower during cloud conditions. Under nighttime cloud droplet conditions, gas-phase NO$_3$ concentrations decreased by about 72 %. The decreased NO$_3$ radical concentration mainly resulted from the increased N$_2$O$_5$ uptake during cloud conditions.

Unfortunately, we did not perform measurements of key radicals during the campaign. However, the simulated maxima of the gas-phase concentrations of OH (C2w: 3.2 × 10$^6$ molecules cm$^{-3}$, C2wo: 3.5 × 10$^6$ molecules cm$^{-3}$) and HO$_2$ (C2w: 2.9 × 10$^8$ molecules cm$^{-3}$, C2wo: 3.8 × 10$^8$ molecules cm$^{-3}$) for Mt. Tai in this study are comparable to the available measurements listed below. Compared with the modeled maximum OH (6.0 × 10$^6$ molecules cm$^{-3}$) and HO$_2$ (7.0 × 10$^8$ molecules cm$^{-3}$)





concentrations at Mt. Tai in 2006 (Kanaya et al., 2009), the OH and HO₂ concentrations reported here are only slightly lower. Moreover, the modeled OH and HO₂ concentrations in this study are lower than those of simulated results over the Chinese megacity Beijing (OH: $9 \times 10^6$ molecules cm⁻³, HO₂: $6.8 \times 10^8$ molecules cm⁻³) (Liu et al., 2012) and much lower than the measured data at a rural site downwind of the megacity Guangzhou, China (OH: $15\text{-}26 \times 10^6$ molecules cm⁻³, HO₂: $3\text{-}25 \times 10^8$ molecules cm⁻³) (Lu et al., 2012). Additionally, the simulated NO₃ radical maxima (C2w: $1.0 \times 10^8$ molecules cm⁻³, C2wo: $1.5 \times 10^8$ molecules cm⁻³) are much lower than those observed at the urban site of Shanghai, China ($2.5 \times 10^9$

molecules cm⁻³) (Wang et al., 2013a). These comparisons suggest that the predicted model data represented a sub-urban oxidant budget along the trajectory above the boundary layer to Mt. Tai.

Similar to the gas-phase, aqueous-phase concentrations of OH and HO₂ also display a distinct diurnal profile with peak concentration around noon, and are largely impacted by cloud interactions. Under daytime cloud droplet conditions, OH aqueous-phase concentrations are increased by a factor of 3, mainly due to the increased direct transfer of OH from the gas

phase. On the other hand, HO₂ aqueous-phase concentrations are decreased by a factor of 0.9 due to aqueous-phase reactions of HO₂ with $Cu^{2+}$ or $Cu^+$. However, higher HO₂ aqueous-phase concentrations appear at the beginning of the daytime cloud. This is caused by the increased H₂O₂ aqueous concentrations due to uptake at the beginning of the daytime cloud period. In the aqueous-phase, H₂O₂ reactions with $Mn^{3+}$ or $FeO^{2+}$ led to a formation of HO₂.

The NO₃ radical presents higher aqueous-phase concentrations during the night, with a minor cloud impact. Under nighttime

cloud conditions, the NO₃ radical concentration decreases by about 13 %. In Table 2, average aqueous-phase concentrations of OH, HO₂ and NO₃ under different conditions are presented. Aqueous-phase NO₃ concentrations are much higher than those of aqueous-phase OH. Considering the normally lower reaction rate constant of aqueous NO₃ radical compared to aqueous OH (Herrmann et al. 2010, Herrmann et al. 2015, Ng et al. 2017), both OH and NO₃ radicals are able to initiate aqueous-phase oxidations under the simulated conditions, contributing to the chemical processing of SOA constituents.

### 3.1.2  Key non-radical oxidants

Figure 2 depicts the modeled gas- and aqueous-phase concentrations of H₂O₂ and O₃ for the C2w and C2wo cases. The corresponding plots for the C4w and C4wo cases can be found in the supplement information because of their similar concentration levels and patterns to C2w and C2wo, respectively (Fig. S2).

Figure 2 shows that, due to active photochemistry, gas-phase concentrations of H₂O₂ and O₃ mainly are increased during the

day and decreased during the night. During cloud periods, H₂O₂ gas-phase concentrations are largely decreased by 96 % due to direct phase transfer and corresponding aqueous-phase chemistry. The predicted cloud impact is minor for gas-phase O₃ concentrations, which is consistent with former studies (Tilgner et al., 2013). During daytime cloud periods, O₃ gas-phase concentrations are decreased by about 4 %. In the C2wo case, H₂O₂ concentrations are substantially higher than those in C2w because of the missing cloud phase sinks (e.g. hydrogen sulfide oxidation). However, O₃ concentrations in C2wo and

C2w are similar because of the minor cloud impact on O₃ in C2w.





In the C2wo case, measured gas-phase $O_3$ concentrations at Mt. Tai ranged from 1.67 to 2.24 × $10^{12}$ molecules $cm^{-3}$ (Fig. 2), which is typical in a Chinese suburban regime (Wang et al., 2013b). However, these concentrations are reached even at the high altitude of Mt. Tai. Additionally, the simulated maxima gas-phase $H_2O_2$ concentrations (C2w: 2.0 × $10^{10}$ molecules $cm^{-3}$, C2wo: 4.7 × $10^{10}$ molecules $cm^{-3}$) are lower than those observed at a rural site downwind of the more polluted area of

Hebei, China (3.0 × $10^{11}$ molecules $cm^{-3}$) (Wang et al., 2016b). The simulated $O_3$ maxima (C2w: 1.8 × $10^{12}$ molecules $cm^{-3}$, C2wo: 2.1 × $10^{12}$ molecules $cm^{-3}$) are slightly lower than those observed at the Nanjing urban area in China (3.6 × $10^{12}$ molecules $cm^{-3}$) (An et al., 2015).

The aqueous-phase $H_2O_2$ presents a similar concentration pattern as the gas-phase $H_2O_2$. Higher $H_2O_2$ aqueous-phase concentrations at the beginning of the daytime cloud are caused by the fast phase-transfer. The subsequent sharp decrease

during the first minute is connected to the efficient hydrogen sulfide oxidation. In the C2wo case, aqueous-phase $O_3$ concentrations increase during the day, and decrease throughout the night. In both daytime and nighttime clouds, $O_3$ aqueous-phase concentrations are increased by about 18 %. The average aqueous-phase concentrations of $H_2O_2$ and $O_3$ in the different cases can be found in Table 2.

## 3.2 Modeled concentrations and formation of key secondary inorganic aerosol constituents

In Fig. 3, modeled concentrations of the most important SIA constituents are plotted, including (i) sulfate (sum of all sulfur compounds with oxidation state +6, mainly $SO_4^{2-}$/$HSO_4^-$), (ii) nitrate (sum of all nitrogen compounds with oxidation state +5, mainly $NO_3^-$), and (iii) ammonium (sum of all nitrogen compounds with oxidation state −3, mainly $NH_4^+$). The corresponding plots for the C4w and C4wo cases are presented in Fig. S3.

*Sulfate*

Conducted field observations together with estimated sulfur oxidation rates using a tracer method in previous studies at Mt. Tai have suggested that sulfate formation is highly related to cloud chemistry (Zhou et al., 2009; Shen et al., 2012; Guo et al., 2012). However, these studies are not able to comprehensively quantify the impact of cloud chemistry on sulfate concentration, and have not performed detailed investigations on chemical formation pathways of sulfate during the transport to Mt. Tai. In this study, we primarily present modeled concentration of sulfate, and discuss the differences between the

different day vs. night and cloud vs. non-cloud cases using a multiphase chemistry model. Moreover, findings of sulfate source and sink chemical reactions are presented for the different model cases.

Figure 3 shows that sulfate concentrations mainly increase under cloud condition throughout the whole simulation due to active in-cloud chemical sulfur oxidation pathways. Although in-cloud residence time is slightly higher during the night, sulfate concentrations increase more in the daytime clouds (35 %) than the nighttime clouds (15 %) because of the increased

aqueous reaction of $HSO_3^-$ with $H_2O_2$ in daytime cloud droplets (Fig. 4). As shown in Fig. 4, about 97 % of sulfate formation fluxes are related to cloud conditions, and mostly occurred in daytime clouds. The aqueous-phase reaction of $HSO_3^-$ with $H_2O_2$ represents the most important sulfate source in daytime cloud with a contribution of 78 %. In the nighttime cloud,





aqueous-phase reaction of $HSO_3^-$ with $H_2O_2$ (42 %), and aqueous reaction of bisulfite with $O_3$ (28 %) are dominated pathways for sulfate formation.

In the C2wo case, sulfate concentrations gradually increase over time (Fig. 3). The highest increase occurs during the day as a consequence of the gas-phase $SO_2$ oxidation by OH (Fig. S4). However, the formed sulfate mass in C2wo case is substantially lower than in the C2w case. At the end of the simulation, the modeled sulfate concentrations are 76.7 and 24.7 μg m$^{-3}$ in the C2w and C2wo cases, respectively. Compared to the maximum (40.0 μg m$^{-3}$), average (32.0 μg m$^{-3}$) and minimum (18.8 μg m$^{-3}$) values of the measured sulfate concentrations at Mt. Tai (Fig. 3), SPACCIM model overestimates

measured concentrations of sulfate in the C2w case, but slightly underestimated the mean sulfate concentration in the C2wo case.

*Nitrate*

Studies at Mt. Tai focused on nitrate suggested that photochemical formation of $HNO_3$ has important contribution to nitrate formation (Zhou et al., 2009; Guo et al., 2012). Wen et al. (2018) found that partitioning of $HNO_3$ from gas to particulate

phase and hydrolysis of $N_2O_5$ is the predominant daytime and nighttime formation pathway of nitrate at Mt. Tai, respectively. However, these studies did not consider nitrate formation processes related to cloud conditions. In this study, we do not only focus on the concentration time profiles of nitrate under day vs. night, but also compared cloud vs. non-cloud cases. Furthermore, we have analyzed nitrate source and sink reactions rates and their contributions in different cases.

As can be seen in Fig. 3, nitrate concentrations are increased throughout the simulation. Under cloud condition, nitrate

concentrations are increased by about 10 % and 24 % during the day and the night, respectively. The concentration time profiles in C2w and C2wo cases show only small differences, indicating that most of the nitrate formation occurs during non-cloud periods. Therefore, the end concentrations of C2w and C2wo do not differ significantly. An analysis of chemical sink and source in the C2w case (Fig. 4) has revealed that nitrate is mainly produced by aqueous-phase $N_2O_5$ hydrolysis during the night. During the day, nitrate is mainly produced by the aqueous-phase reaction of $HNO_4$ and $HSO_3^-$, gas-phase

reaction of OH and $NO_2$, and aqueous-phase $N_2O_5$ hydrolysis.

A comparison of daytime and nighttime fluxes in the C2wo case has revealed that 31 % and 69 % of nitrate formation fluxes occur at day and at night, respectively. In the C2wo case, nighttime nitrate is mainly produced by aqueous-phase $N_2O_5$ hydrolysis with a contribution of 92% (Fig. S4), which agrees with the result in Wen et al. (2018). However, gas-phase reaction of OH + $NO_2$ to gaseous $HNO_3$ and further partitioning into the aerosols is the most important source for daytime

nitrate with a contribution of 56% (Fig. S4). Aqueous-phase $N_2O_5$ hydrolysis is also significant for daytime nitrate formation with a contribution of 30%. In comparison, Wen et al. (2018) have suggested the partitioning of $HNO_3$ from gas to the particulate phase to be the most important formation pathway for daytime nitrate with a contribution of 94%. The $N_2O_5$ hydrolysis has been identified as the second most important pathway with a contribution of 4-6 %.

The modeled nitrate concentrations are 69.5 and 65.3 μg m$^{-3}$ at 96 h in the C2w and C2wo cases, respectively. Compared to

the maximum (25.0 μg m$^{-3}$), average (14.0 μg m$^{-3}$) and minimum (6.8 μg m$^{-3}$) values of the measured nitrate concentrations



at Mt. Tai (Fig. 3), the model simulations overestimate the measured concentrations in both cases. Potential reasons are discussed below.

*Ammonium*

Measured ammonium concentrations at Mt. Tai can be strongly impacted by acidification and cloud chemistry (Guo et al.,
2012; Li et al., 2017a). Still, a detailed analysis of the occurring processes is missing. Therefore, we provide a detailed insight into the ammonium concentration variation trends, and the impact of acidification and cloud processing along the simulated trajectories to Mt. Tai.

Similar to sulfate and nitrate, ammonium concentrations also gradually increased throughout the simulation due to the included emissions rates and the followed uptake of gaseous $NH_3$ into the aqueous phase. Ammonium concentrations raised
in cloud droplets both during the day and night, because a more efficient uptake into the larger cloud water volume. After cloud evaporation, ammonium aerosol concentrations are increased due to stronger partitioning into more acidified deliquesced particles. In the C2wo case, ammonium concentrations are also increased both during the day and night. However, modeled aerosol mass of ammonium in the C2wo case is lower than that in the simulation case with cloud interaction (C2w case). In the C2wo case, less sulfate is formed. Consequently, deliquesced aerosol particles are less
acidified and a larger fraction of ammonium stays in the gas phase as gaseous $NH_3$. The simulated ammonium concentrations after 96 h are 42.7 and 25.9 µg m$^{-3}$ in the C2w and C2wo cases, respectively, which are higher than the measured concentrations (maximum: 18.0 µg m$^{-3}$, average: 15.6 µg m$^{-3}$, minimum: 7.9 µg m$^{-3}$).

The differences between the modeled and measured concentrations of sulfate, nitrate and ammonium can be attributed to several issues such as (i) the indefiniteness of the input emission data, (ii) the initial concentrations, (iii) the missing
entrainment/detrainment processes, and (iv) the performed heating of the inlet during the sampling of wet aerosol (see Sect. 3.3.2 for further details).

### 3.3 Modeled concentrations of DCRCs and comparison with field data

In recent years, a number of field observations on DCRCs have been conducted in the NCP. For example, He et al. (2013), Ho et al. (2015), Zhao et al. (2018) and Yu et al. (2019) have observed DCRCs in Beijing; Wang et al. (2009), Kawamura et
al. (2013), Meng et al. (2018) and Zhao et al. (2019) have measured DCRCs at Mt. Tai. Our field observation about DCRCs at Mt. Tai has been reported in Zhu et al. (2018). However, these studies are focused on DCRC concentrations, molecular compositions, temporal variations, size distributions, source implications and stable carbon isotopic composition. They have not investigated the chemical formation of DCRC concentrations along the trajectory as well as the impact of cloud and non-cloud conditions on DCRC concentrations. To our knowledge, a multiphase chemical model study investigating the DCRCs
concentration variations and their chemical processing along the trajectory to Mt. Tai considering day/night and cloud/non-cloud cases has not been yet reported.





### 3.3.1 Modeled concentrations of DCRCs

Figure 5 shows the modeled aqueous-phase concentrations of Gly, $\omega C_2$, $C_2$, methylglyoxal (MGly), Pyr, and malonic acid ($C_3$) both in the C2w and C2wo cases as well as the values measured at Mt. Tai.

*Dicarbonyl compounds*

In the C2w case, Gly and MGly concentration patterns shows a substantial uptake into cloud droplets. Gly concentrations decreased during the daytime and nighttime cloud droplet periods due to in-cloud oxidation processes. On the other hand, MGly concentrations display a decrease in the daytime cloud droplets, but an increase under nighttime cloud conditions. This might have been caused by the fact that the aqueous oxidation fluxes under nighttime cloud conditions are lower than

the ones under daytime. In the C2wo case, Gly and MGly concentrations are very low due to the low partitioning towards aqueous particles that has been predicted by the model. The effect of a potentially higher partitioning constant of Gly (Volkamer et al., 2009; Ip et al., 2009) is investigated in Sect. 3.5.3.

*$C_2$ carboxylic acids*

In the C2w case, aqueous-phase concentrations of $\omega C_2$ are increased under both daytime and nighttime cloud conditions and

early in the night of non-cloud periods. On the other hand, $\omega C_2$ concentrations are lowered during the day and later in the night under non-cloud conditions. In the C2wo case, $\omega C_2$ is decreased during the morning periods but is increased in the late afternoon and at night.

In the C2w case, modeled aqueous-phase concentrations of $C_2$ are increased under daytime cloud conditions and daytime aqueous particle conditions, but are lowered during nighttime cloud periods and under nighttime aqueous particle conditions.

In the C2wo case, $C_2$ is increased during the day but is decreased during the night.

*$C_3$ carboxylic acids*

In the C2w case, Pyr concentrations are raised during the daytime and nighttime cloud conditions as well as in the late mornings of the non-cloud periods. Pyr concentrations are decreased in the early morning, afternoon and nighttime non-cloud conditions (caused by the efficient degradation from the reaction with aqueous-phase $H_2O_2$). In the C2wo case,

aqueous-phase concentration profile of Pyr is characterized by an increase during the morning and early afternoon period and by a strong decrease during the late afternoon and night. Pyr shows a high correlation with aqueous-phase $H_2O_2$ in the C2wo case due to its efficient $H_2O_2$ decay (Fig. S5).

The aqueous-phase $C_3$ concentrations are increased during cloud formation due to the uptake of gaseous $C_3$ into cloud droplets. Moreover, $C_3$ concentrations are increased during non-cloud periods. In the C2wo case, $C_3$ concentrations are

increased slightly during the night but even less during the day. In comparison to the C2wo case, the C2w case shows higher concentrations of $C_3$, indicating that cloud oxidation processes are very important for $C_3$ aqueous-phase formation under the simulated conditions. The production of $C_3$ is 27 % higher in the C2w case than in the C2wo case.

In the C4w and C4wo cases, the modeled Gly, $\omega C_2$, $C_2$, MGly, Pyr, and $C_3$ concentrations show similar patterns to those in the C2w and C2wo cases, respectively (Fig. S6).



### 3.3.2 Comparison to field observations

The ratios of the average concentration of modeled and measured DCRCs can be found in Table 3. The results show that model predictions are higher than the measured concentrations of $C_3$, Pyr and $\omega C_2$ in both C2w and C2wo cases. Moreover, the concentration ratios of Pyr and $\omega C_2$ in the C2w case are much higher than in the C2wo case. On the other hand, model results underpredict the $C_2$, Gly and MGly concentrations in both cases.

SPACCIM overestimates the measured $\omega C_2$ concentrations, but underestimates the measured $C_2$ ones, suggesting the conversion of $\omega C_2$ might be implemented less efficiently into CAPRAM. The partitioning treatment of MGly may be not sufficient enough to predict the measured MGly aerosol concentrations in the field because of model simplicity. Kroll et al. (2017) have found that a possible hydration of MGly in the gas-phase might lead to an enhanced uptake into aqueous particles. Thus, maybe the MGly uptake is underestimated. Additionally, other important processes that can effectively enhance partitioning of MGly are not yet considered, such as reversible oligomerizations (Ervens and Volkamer, 2010). As a result, the modeled aqueous-phase MGly concentration is rather low and about three orders of magnitude lower than the measured data. Based on this finding, a sensitivity study has been performed (see Sect. 3.5 for details).

The over- and underestimation of the measured concentrations of inorganic and organic aerosol constituents could have the following reasons:

(1) Input data: Indefiniteness of emission data and initial concentrations. The emission data are obtained through model calculations, not field measurements. The height of Mt. Tai (about 1500 m) also causes its input to be indefinite. Furthermore, initial concentrations in originated areas are obtained through related references rather than field measurements, which also lead to indefiniteness.

(2) Field measurement: $PM_{2.5}$ samples are the only ones available, so a possible contribution of larger particles might have been missed. Moreover, heating the inlet during the sampling of wet aerosol has definitely lowered the measured concentrations of more volatile compounds such as ammonium nitrate.

(3) MCM mechanism: Some species' crucial gaseous precursors are efficiently destroyed by MCM. For example, it is recommended that gas-phase oxidation of acetic acid by OH proceed via both an H-abstraction from the OH-group and the $CH_3$-group (Sun et al., 2009). Nonetheless, only an H-abstraction for OH-group is implemented in the MCM. This oxidation scheme is implemented for all carboxylic acids. The disadvantage of the MCM probably leads to an underestimation of the functionalized acids and need to be resolved in more detail. However, the development of an improved gas-phase acid oxidation scheme for the MCM goes beyond the scope of this study.

(4) CAPRAM mechanism: Missing sources of selected DCRCs from higher organic chemistry, such as the aqueous-phase chemistry of aromatic compounds (Hoffmann et al. 2018).

(5) SPACCIM model: The model neglects non-ideal solution effects and do not consider organic salt formation. These factors possibly result in overestimated or underestimated results. Recent studies by Rusumdar et al. (2019) have demonstrated that non-ideal chemistry treatment led to much higher concentrations of $C_2$ and $\omega C_2$.





Apart from MGly, the concentration ratios of the modeled and measured species ranged from 0.1 to 8.3, which can be regarded as an acceptable range due to the model and input data limitations. The SPACCIM model with an implemented
MCM3.2/CAPRAM4.0 is a numerical tool that can help us understand the complexity of the multiphase processing of DCRCs better. However, the present study also demonstrates that there are still huge gaps in knowledge about the formation and degradation of secondary aerosol constituents. Hence, further laboratory investigations and modeling work are necessary.

### 3.4   Model investigations of chemical sources and sinks of selected DCRCs

Although field observations have speculated about several potential formation pathways of some DCRCs species by correlations or ratios analyses (Hegde and Kawamura, 2012; Kawamura et al., 2013; Zhao et al., 2019), the detailed pathways of DCRCs need to be studied.

Multiphase model simulations are a suitable tool to investigate DCRCs formation processes. In recent years, DCRCs formation processes have been examined by several model studies. For example, Tilgner and Herrmann (2010) have
modeled gas and aqueous phases processing of $C_2$-$C_4$ carbonyl compounds and carboxylic acids by SPACCIM; Ervens et al. (2004) have discussed formation pathways of Pyr and $C_2$-$C_6$ dicarboxylic acids in gas and aqueous phases; Lim et al. (2005) have reported the formation pathways of Gly, MGly, Pyr and $C_2$ by isoprene oxidation in gas and aqueous phases using a photochemical box model; Huisman et al. (2011) have investigated the driving processes of Gly chemistry by Master Chemical Mechanism (MCM, v3.1). These studies have suggested that DCRC formations are related to the oxidations of
anthropogenic (e.g. toluene and ethylene) and biogenic (e.g. isoprene) gas-phase VOCs precursors. The emissions of these anthropogenic and biogenic VOC in China are much higher than those reported in in above references (Sindelarova et al., 2014; Bauwens et al., 2016). However, multiphase model simulations are sparsely used to study DCRCs formations in China. Therefore, the present study aimed at a detailed investigation of the formation pathways of selected DCRCs under day vs. night and cloud vs. non-cloud cases along the trajectories approaching to Mt. Tai.
Due to the similar concentration levels and corresponding variation trends of $\omega C_2$, $C_2$, Pyr and $C_3$ in clusters 2 and 4, the source and sink flux analyses are only presented and discussed for the C2 case. Additionally, the corresponding plots of the four compounds in the C2wo case are presented in the supplement information (Fig. S7).

### 3.4.1   Glyoxylic acid ($\omega C_2$)

In Fig. 6, the multiphase source and sink fluxes of $\omega C_2$ (C2w case) on the third model day are plotted. The results reveal a
net formation flux that mainly occurs during cloud conditions as well as a net degradation during non-cloud periods. About 94 % of the net formation flux of $\omega C_2$ is simulated under cloud condition. However, the non-cloud conditions represent 100 % of the net sink flux of $\omega C_2$.

Under daytime and nighttime cloud conditions, the major formation pathways of $\omega C_2$ are aqueous-phase reactions of hydrated Gly with the OH and $SO_x^-$ radical (contribution: 60 % at day, 86 % at night), which is similar with results in Tilgner





and Herrmann (2010) and Tilgner et al. (2013). The aqueous-phase oxidation of glycolate by OH is also significant under daytime cloud conditions with a contribution of 18 %. Under daytime and nighttime non-cloud conditions, aqueous reactions of hydrated Gly (contribution: day: 20 %, night: 20 %) and gas-phase decay of 3-methyl-6-nitrocatechol (contribution: day: 14 %, night: 24 %) are significant for $\omega C_2$ formation. Other reactions contributed less than 5 % to the overall source flux.

Under daytime clouds, $\omega C_2$ sink is dominated by aqueous-phase reaction of hydrated glyoxylate with OH (contribution:

88 %), which is consistent with Ervens et al. (2004). During nighttime clouds, however, aqueous-phase reactions of hydrated glyoxylate with $NO_3$ (45 %) and OH (28 %) are the most important sinks. In contrast to those under cloud conditions, gas-phase $\omega C_2$ photolysis (57 %) and gas-phase reaction of $\omega C_2$ with OH (18 %) acts as key sinks of $\omega C_2$ under daytime non-cloud conditions. Under nighttime non-cloud conditions, the sink fluxes of $\omega C_2$ are low and therefore unimportant.

### 3.4.2 Oxalic acid ($C_2$)

Figure 6 also depicts the source and sink fluxes of $C_2$ in the C2w case. The model has simulated a net formation flux during the non-cloud periods and a net degradation in the early morning hours when non-clouds are present but the photolysis of iron-oxalate complexes is efficient. A net formation of about 94 % $C_2$ is simulated under non-cloud conditions. About 74 % of the net $C_2$ sink fluxes are predicted during the early morning non-cloud period, and 26 % are related to the cloud oxidation fluxes.

The most important source of $C_2$ in the aqueous phase under cloud condition is aqueous oxidations of hydrated glyoxylate with the OH radical, which agrees with other model results (Tilgner and Herrmann, 2010; Ervens et al., 2004). This formation pathway contributes to $C_2$ formation with about 72 % during the day and 87 % during the night. In contrast to that under cloud condition, the most important $C_2$ formation pathway during daytime and nighttime non-cloud conditions is the aqueous-phase oxidation of hydrated $\omega C_2$ with the OH radical (contribution: day: 39 %, night: 52 %). The field observations

also suggested that aqueous-phase oxidation of $\omega C_2$ is the most important formation pathway of $C_2$ (Kundu et al., 2010; Kawamura et al., 2013; He and Kawamura, 2010), but they are not able to quantify contribution and the responsible specific oxidation pathways. Other reactions contribute less than 5 % to the overall source flux.

The most important sink of $C_2$ under daytime cloud conditions is the photolysis of iron-oxalate complexes, with a contribution of about 93 %. The result is similar to reported laboratory experiment findings (Pavuluri and Kawamura, 2012)

and aqueous model simulation (Tilgner and Herrmann, 2010). On the other hand, aqueous reaction of oxalate with $NO_3$ (80 %) is the most important sink in the nighttime cloud case. Under nighttime non-cloud conditions, $C_2$ sink is dominated by the reaction of the oxalate monoanion with $NO_3$ (81 %). Under daytime non-cloud conditions, only significant $C_2$ sink is the photolysis of iron-oxalate complexes. However, photolysis of iron-oxalate complexes under aqueous particle conditions is most likely overestimated in the present SPACCIM model studies. Recent studies by Rusumdar et al. (2019) – using an

advanced SPACCIM model with a non-ideality treatment, but with a more reduced chemical CAPRAM scheme – reveal that the formation and consequently the photolysis of iron-oxalate complexes is inhibited under aqueous particle conditions. The





possible overestimation of the photolytic decay leads to a significantly low $C_2$ concentration and might be thus the reason for the underestimated $C_2$ concentration.

### 3.4.3 Pyruvic acid (Pyr)

The modeled source and sink fluxes of Pyr in the C2w case on the third model day can be found in Fig. 6. A net formation flux is modeled mainly under cloud condition, especially during the day, along with a net degradation during non-cloud periods. About 72% of the net Pyr flux occurs in clouds, whereas 28 % is formed under non-cloud conditions. However, 100 % of the multiphase Pyr net sink fluxes are related to non-cloud oxidation.

Under cloud condition, the dominant source for Pyr is hydrolysis of the aqueous-phase oxidation product of nitro
2-oxopropanoate, with a contribution of 89 % during the day and 70 % during the night. The result is different from former model studies, e.g. Ervens et al. (2004), Lim et al. (2005), Tilgner and Herrmann (2010) and Tilgner et al. (2013), which modeled the aqueous oxidations of MGly as the major formation pathway of Pyr. However, these model studies have also modeled different environmental conditions with much lower anthropogenic pollution including lower $NO_x$ and $NO_3$ radical budgets compared to the Chinese conditions. Furthermore, the former studies have also used rather simple gas-phase
mechanisms and lacks potential production pathways from higher organic compounds. Similarly, the aqueous oxidation of nitro 2-oxopropanoate is identified as a major source under non-cloud conditions, with a contribution of 87 % during the day and 74 % during the night.

The key sinks of Pyr under daytime cloud conditions are aqueous-phase reactions of pyruvate with OH (58 %) and $H_2O_2$ (29 %). This finding is consistent with results in laboratory experiment (Carlton et al., 2006). Under night cloud conditions,
the sink fluxes are very low and therefore can be ignored. Under daytime and nighttime non-cloud conditions, dominant sinks are aqueous-phase reactions of pyruvate with $H_2O_2$ (contribution: day: 57 %, night: 72 %) and free Pyr with $H_2O_2$ (contribution: day: 13 %, night: 15 %). Additionally, gas-phase Pyr photolysis (15 %) is also important under daytime non-cloud conditions.

### 3.4.4 Malonic acid (C₃)

In Fig. 6, the modeled source and sink fluxes of $C_3$ in the C2w case are presented for the third model day. A net formation flux can be seen under daytime cloud conditions and both daytime and nighttime non-cloud conditions. A net degradation is only found under nighttime cloud conditions. A $C_3$ net formation flux is about 82 % under non-cloud conditions and 18 % under daytime cloud conditions.

The major modeled sources of $C_3$ under daytime and nighttime cloud conditions are aqueous oxidation reactions of hydrated
3-oxopropanoic acid (contribution: day: 48 %, night: 50 %) and hydrated 3-oxopropanoate (contribution: day: 45 %, night: 47 %). However, under non-cloud conditions, aqueous-phase oxidation of hydrated 3-oxopropanoic acid is dominant, with a contribution of 79 % during the day and 88 % during the night.





Differences between the sink fluxes under cloud and non-cloud conditions are modeled. The $C_3$ sinks under daytime clouds are dominated by aqueous-phase reaction of malonate monoanion with OH. Its contribution to total sink flux under the daytime cloud is 70 %. Contrary to that, aqueous-phase reaction of malonate monoanion with $NO_3$ is the most important sink under nighttime cloud conditions with a contribution of 75 %. The predominated sink pathway of $C_3$ in daytime cloud is consistent with Ervens et al. (2004), but that in nighttime cloud is different due to the missing $NO_3$ radical pathways in their mechanism. The modeled $C_3$ sinks under non-cloud conditions are much lower than the sinks under cloud conditions, and thus are unimportant.

### 3.5 Sensitivity studies

Due to the similarity between clusters 2 and 4, as mentioned above, sensitivity tests are only performed under cluster 2 conditions. The present study investigated (i) impact of emissions on modeled compound concentrations, (ii) key precursors of DCRCs, and (iii) impact of increased Gly aerosol partitioning on Gly, $\omega C_2$ and $C_2$.

#### 3.5.1 Impact of emissions

First, sensitivity tests are performed to evaluate the effect of different emission strengths on the concentrations of key secondary inorganic compounds and selected DCRCs during the transport. The emission sensitivities of sulfate, nitrate, ammonium, Gly, $\omega C_2$, $C_2$, MGly, Pyr and $C_3$ are investigated by doubling the applied emission fluxes of the base case. The results of the sensitivity tests can be found in Fig. 7. The modeled concentrations of sulfate, nitrate, ammonium, Gly, $\omega C_2$, $C_2$, MGly, Pyr and $C_3$ are increased by a factor of about two when the emissions doubled, which suggests an almost linear dependency. The results indicate that the modeled concentrations of secondary aerosol are very sensitive to the emissions in the model.

#### 3.5.2 Identification of the key precursors of $C_2$, Pyr and $\omega C_2$

Further sensitivity tests are conducted to identify key primary precursors of DCRCs during atmospheric transport. We have adopted the relative incremental reactivity (RIR) method by Carter and Atkinson (1989) for the sensitivity tests. The RIR method has been applied already in a former study to investigate the precursors of peroxy acetyl nitrate in urban plume in Beijing (Xue et al., 2014).

As can be seen in Fig. 8, $C_2$ formation in the C2w case is mostly sensitive to aromatics and alkenes. Among the aromatic compounds, toluene is the most important one for $C_2$ formation. However, other aromatic species (such as xylene, ethyl benzene, isopropyl benzene) present negative RIRs. Among the alkenes, isoprene and 1,3-butadiene are dominant, but ethene shows negative RIR. The alkane RIRs are all negative. Positive and negative RIRs probably depend on oxidant variations. As shown in Fig. S8, the important sources of $C_2$ in the C2w case are the oxidation of hydrated $\omega C_2$ by the OH radical and sulfur containing radicals ($SO_x^-$). A reduced concentration ratio of the OH or $SO_x^-$ is calculated in case of a 20 % decrease of emission data. After the calculation, a correlation with RIR values has been performed. Fig. 9 shows that OH and $SO_x^-$





radicals have high and moderate correlations with $C_2$-RIRs in C2w case, respectively, suggesting that the concentration
variations of OH and $SO_x^-$ radicals are the reason for the positive and negative $C_2$-RIRs in the C2w case.

In the C2wo case, alkenes account for the highest RIR. The RIR of alkenes is more than two times higher than that of the second highest group (aromatic compounds). Among the alkenes, the dominant compound is isoprene. Contrary to the C2w case, 1,3-butadiene reveals very low RIR under C2wo condition. In the C2wo case, ethene exhibits a positive but low RIR. Among aromatic compounds, toluene shows the highest RIR. Xylene, ethyl benzene and isopropyl benzene also presents significantly positive RIR values in the C2wo case. Alkanes have negative RIRs again. As shown in Fig. S8, in the C2wo case, the reactions of dissolved $\omega C_2$ with OH radicals represent the major pathways. Strong correlations between $C_2$-RIRs and OH radical in the C2wo case (Fig. 9) indicate that the calculated positive and negative RIRs in the C2wo case are due to OH radical concentration variations.

As for Pyr, in both C2w and C2wo cases, alkenes are the dominant precursor group with the largest RIRs. The major compound is isoprene. The absolute RIR values for other selected species are less than 0.05. These results indicated that Pyr formation during atmospheric transport is highly sensitive to isoprene.

In the C2w case, aromatic compounds are the most significant precursors of $\omega C_2$ with high positive RIR. However, individual aromatic species listed in Fig. 8 are characterized by negative RIRs. Alkanes and alkenes show negative values. However, isoprene and 1,3-butadiene have positive RIRs, and their high levels suggests that they are key species controlling $\omega C_2$ formation during the modeled summer conditions. As shown in Fig. S8, oxidations of dissolved Gly by OH and $SO_x^-$ radicals are the most important sources for $\omega C_2$ formation in the C2w case. High correlations between OH and $SO_x^-$ radicals with $\omega C_2$-RIR values (Fig. 9) suggest that positive and negative $\omega C_2$-RIRs in the C2w case are a result of variations of the two oxidants.

Figure 8 shows that aromatic compounds account for the highest RIR under C2wo condition, and toluene is a major contributor. Ethyl benzene and isopropyl benzene also made significant contributions. The alkene RIRs are the next highest. Isoprene is the most abundant compound during $\omega C_2$ formation. As in cases of $C_2$ and Pyr, alkanes also have negative $\omega C_2$-RIRs. $\omega C_2$ production mainly depends on the oxidation of dissolved Gly by OH radical in the C2wo case, and variation trend of OH radical are the reason for the positive and negative $\omega C_2$-RIRs (see Fig. S8 and Fig. 9).

### 3.5.3  Identifying the impact of increased Gly aerosol partitioning

Phase partitioning between gas and aqueous phase in a multiphase model can be affected, e.g. by salting-in/salting-out effects and other reversible accretion reactions (Herrmann et al., 2015). For example, Ip et al. (2009) and Kampf et al. (2013) have reported that $SO_4^{2-}$ and ammonium sulfate can have a significant effect on the uptake of Gly into an aqueous solution. Therefore, a sensitivity study considering increased Gly aerosol partitioning has been done to evaluate the changes of Gly, $\omega C_2$ and $C_2$ concentrations. This has been realised by increasing the CAPRAM Gly Henry's law constant (1.4 mol $l^{-1}$ $atm^{-1}$, Betterton and Hoffmann, 1988). An increased Gly Henry's law constant (raised by two orders of magnitude) have been applied, which is close to the value reported in Volkamer et al. (2009). As can be seen in Fig. 10, compared to the base



C2wo case performed without an increased Gly Henry's law constant, the modeled $\omega C_2$ and $C_2$ aerosol concentrations increase by three and two times, respectively. This result suggested that an increased Gly aerosol partitioning might play an important role in $\omega C_2$ and $C_2$ aqueous-phase formation.

## 545  4    Conclusions

The present study focuses on the formation processes of secondary aerosols constituents along trajectories to Mt. Tai using the multiphase chemistry air parcel model SPACCIM. The modeled radical and non-radical concentrations (such as gas-phase OH concentration: $3.2 \times 10^6$ molecules cm$^{-3}$ in C2w and $3.5 \times 10^6$ molecules cm$^{-3}$ in C2wo) suggest that the atmospheric environment of Mt. Tai (~ 1.5 km a.m.s.l.) is still characterized by a sub-urban oxidants budget at the altitude of
about 1.5 km. Compared to previous studies at Mt. Tai, this study is the first that investigates the formation processes of secondary aerosols constituents along different trajectories to Mt. Tai under day vs. night and cloud vs. non-cloud conditions in detail. The aqueous reaction of $HSO_3^-$ with $H_2O_2$ has been identified as the major contributor to $SO_4^{2-}$ formation (contribution: 67 % in C2w, 60 % in C2wo). $NO_3^-$ formation is higher during the night than during the day. The major pathways are aqueous-phase $N_2O_5$ hydrolysis (contribution: 72 % in C2w, 70 % in C2wo) and gas-phase reaction of OH +
$NO_2$ (contribution: 18 % in C2w, 21 % in C2wo). Aqueous-phase reactions of hydrated Gly, hydrated $\omega C_2$, nitro 2-oxopropanoate and hydrated 3-oxopropanoic acid are dominant formation pathways of $\omega C_2$, $C_2$, Pyr and $C_3$, respectively. Sensitivity tests indicate isoprene, 1,3-butadiene and toluene as key precursors of $\omega C_2$ and $C_2$. The model data analyses show that isoprene is the predominant precursor for Pyr. When emissions are doubled, the modeled SOA compound concentrations increase about two times, suggesting that gaseous VOC emissions are a driving factor for the modeled SOA compound
concentrations. The results indicate the importance of further emission reduction efforts to gain a better air quality in this part of China.

Additionally, the simulations show that an increased Gly aerosol partitioning plays an important role in $\omega C_2$ and $C_2$ aqueous-phase formation. Finally, the present study reveals that, in order to better understand the presence, formation, chemical fate and phase partitioning of DCRCs in the troposphere in the future, comprehensive aerosol and cloud field studies, advanced
mechanistic laboratory studies and more chemical processes model studies are necessary. In case of field investigations, advanced measurements characterizing chemical gas and aerosol compositions with a high time resolution are needed to enable better comparison with and evaluations of present multiphase models. On the other hand, for future model comparisons also more advanced models are required. Those should include a more detailed chemistry description, a detailed treatment of non-ideal solution effects, and an improved treatment of the phase-partitioning of organic compounds, e.g.
considering salting-in/salting-out effects and other reversible accretion reactions. Thus, the obeserved differences between modelled data and measurements could be fixed.





**Data availability**

The data used in this study are available from the corresponding author upon request (email: herrmann@tropos.de,
xuelikun@sdu.edu.cn).

**Supplement**

The supplement related to this article is available online at:

**Author contributions**

YHZ, AT and HH designed the SPACCIM modeling work. YHZ, AT and EHH performed the different SPACCIM
simulations. YHZ, AT, EHH and HH analysed the SPACCIM simulation results. YHZ and LKX performed and interpreted
the RIR analysis. YHZ, AT, KK, LXY and WXW compared the model results with field data. YHZ, AT, EHH, HH and
LKX wrote the paper, and prepared the manuscript material with contributions from all the co-authors.

**Competing interests**

The authors declare that they have no conflict of interest.

**Acknowledgements**

The authors acknowledge the financial support from the National Key Research and Development Program of China
(No.: 2016YFC0200500), the National Natural Science Foundation of China (No.: 21577079, 41922051) and the Japan
Society for the Promotion of Science through Grant-in-Aid (No.: 24221001). We thank the European Commission for
support of the MARSU project under contract 69089. The authors also acknowledge the China Scholarship Council for
supporting Yanhong Zhu to study in the project at the Atmospheric Chemistry Department (ACD) of the Leibniz Institute for
Tropospheric Research (TROPOS), Germany.



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



**Tables**

**Table 1. Acronyms of the performed model simulations.**

| Acronym | Acronym description |
|---------|---------------------|
| C2w | Cluster 2 with cloud interaction |
| C2wo | Cluster 2 without cloud interaction |
| C4w | Cluster 4 with cloud interaction |
| C4wo | Cluster 4 without cloud interaction |


**Table 2. Average aqueous-phase concentrations (mol l⁻¹) of modeled radical and non-radical oxidants in different simulations.**

| Oxidant | C2w | | | | C2wo | |
|---------|-----|-----|-----|-----|------|------|
| | day cloud | night cloud | day particle | night particle | day particle | night particle |
| $OH_{(aq)}$ | $9.6 \times 10^{-14}$ | $6.1 \times 10^{-15}$ | $1.8 \times 10^{-14}$ | $3.0 \times 10^{-15}$ | $3.0 \times 10^{-14}$ | $8.9 \times 10^{-15}$ |
| $HO_{2(aq)}$ | $2.1 \times 10^{-9}$ | $6.2 \times 10^{-10}$ | $3.5 \times 10^{-9}$ | $1.6 \times 10^{-9}$ | $3.0 \times 10^{-9}$ | $1.3 \times 10^{-9}$ |
| $NO_{3(aq)}$ | $1.0 \times 10^{-13}$ | $5.0 \times 10^{-13}$ | $8.8 \times 10^{-14}$ | $5.6 \times 10^{-13}$ | $9.4 \times 10^{-14}$ | $6.9 \times 10^{-13}$ |
| $H_2O_{2(aq)}$ | $7.6 \times 10^{-6}$ | $1.0 \times 10^{-5}$ | $7.7 \times 10^{-5}$ | $6.1 \times 10^{-5}$ | $2.4 \times 10^{-4}$ | $2.2 \times 10^{-4}$ |
| $O_{3(aq)}$ | $1.1 \times 10^{-9}$ | $1.0 \times 10^{-9}$ | $8.7 \times 10^{-10}$ | $8.5 \times 10^{-10}$ | $1.0 \times 10^{-9}$ | $1.0 \times 10^{-9}$ |

**Table 3. Ratios of the average concentrations of the modeled and measured DCRC compounds in the different model trajectories at Mt. Tai.**

| Compound | Model case | | | |
|----------|------------|------|------|------|
| | C2w | C2wo | C4w | C4wo |
| $C_2$ | 0.3 | 0.2 | 0.3 | 0.3 |
| $\omega C_2$ | 7.4 | 2.7 | 8.0 | 2.9 |
| $C_3$ | 1.6 | 1.3 | 1.9 | 1.5 |
| Pyr | 8.3 | 3.0 | 8.1 | 2.6 |
| Gly | 0.1 | 0.1 | 0.1 | 0.1 |
| MGly | 1.8E-3 | 1.1E-3 | 2.2E-3 | 1.7E-3 |





**Figures**


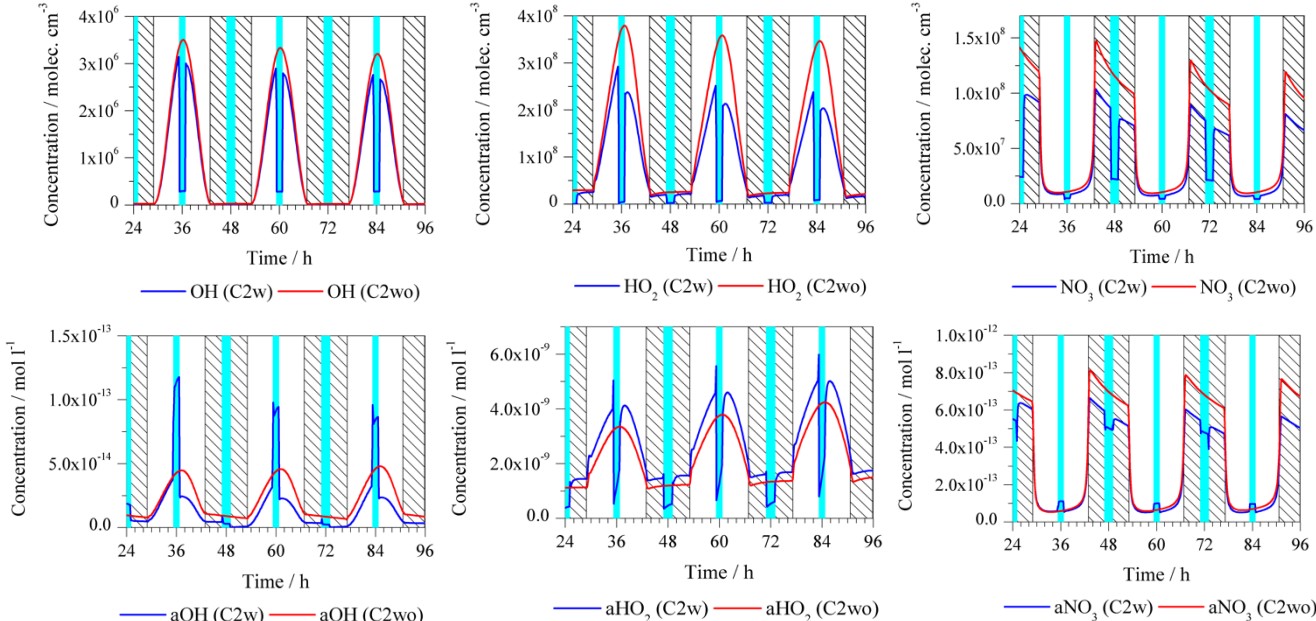

**Figure 1. Time series of the modeled gas- (above: molec. cm⁻³) and aqueous- (below: mol l⁻¹) phase radical oxidant concentrations of the C2w and C2wo cases, respectively (light blue column: cloud; shadow: night; a: aqueous phase). For acronyms, see Table 1.**






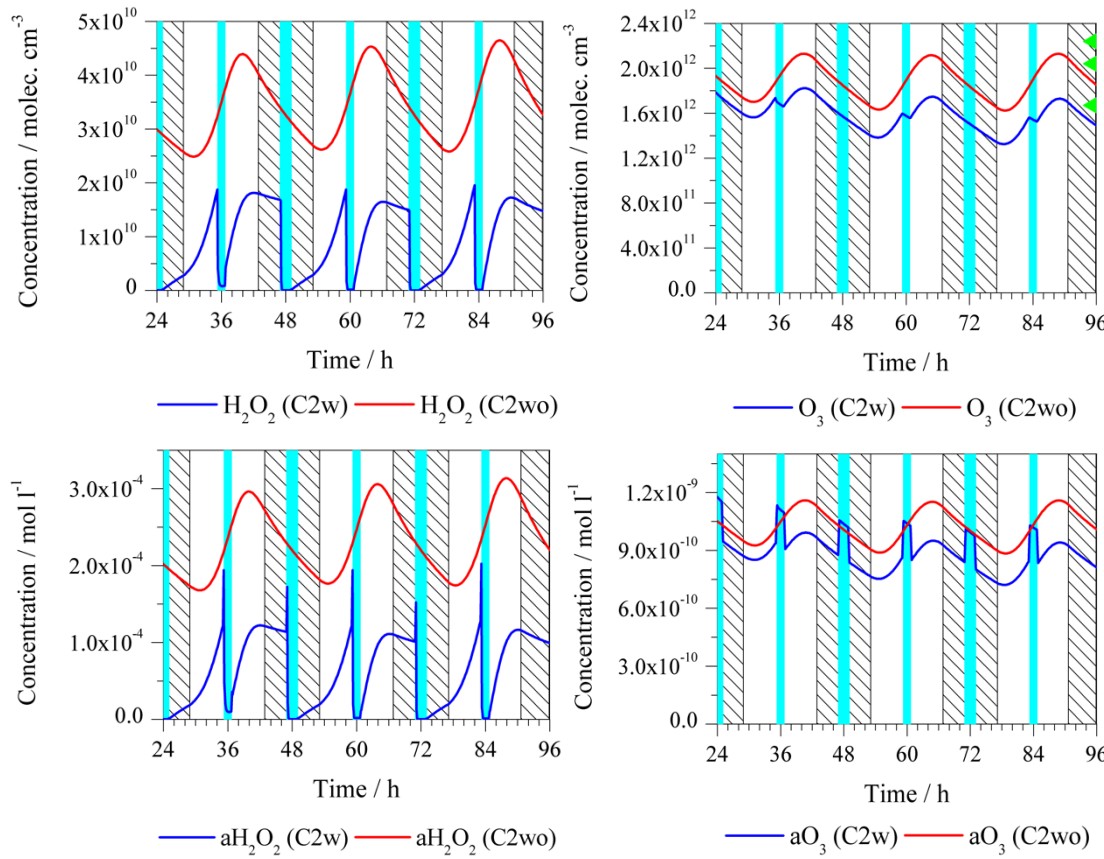

**Figure 2. Time series of modeled gas- (above: molec. cm$^{-3}$) and aqueous- (below: mol l$^{-1}$) phase non-radical oxidant concentrations in the C2w and C2wo cases (light blue column: cloud; shadow: night; a: aqueous phase; green triangle: the maximum (above), average (middle) and minimum (below) value of measured concentration at Mt. Tai).**




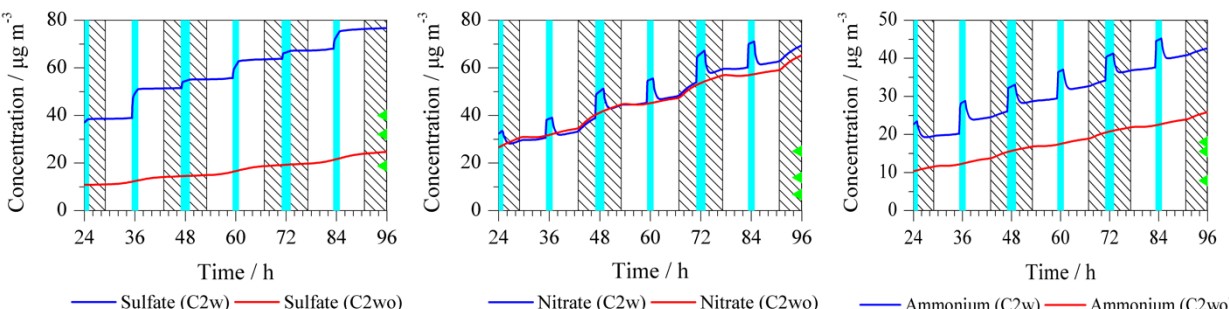

**Figure 3. Time series of the modeled aerosol mass concentrations (µg m⁻³) of key secondary inorganic aerosol constituents (sulfate, nitrate and ammonium in the C2w and C2wo cases (light blue column: cloud; shadow: night; green triangle: the maximum (above), average (middle) and minimum (below) value of measured concentration at Mt. Tai)).**



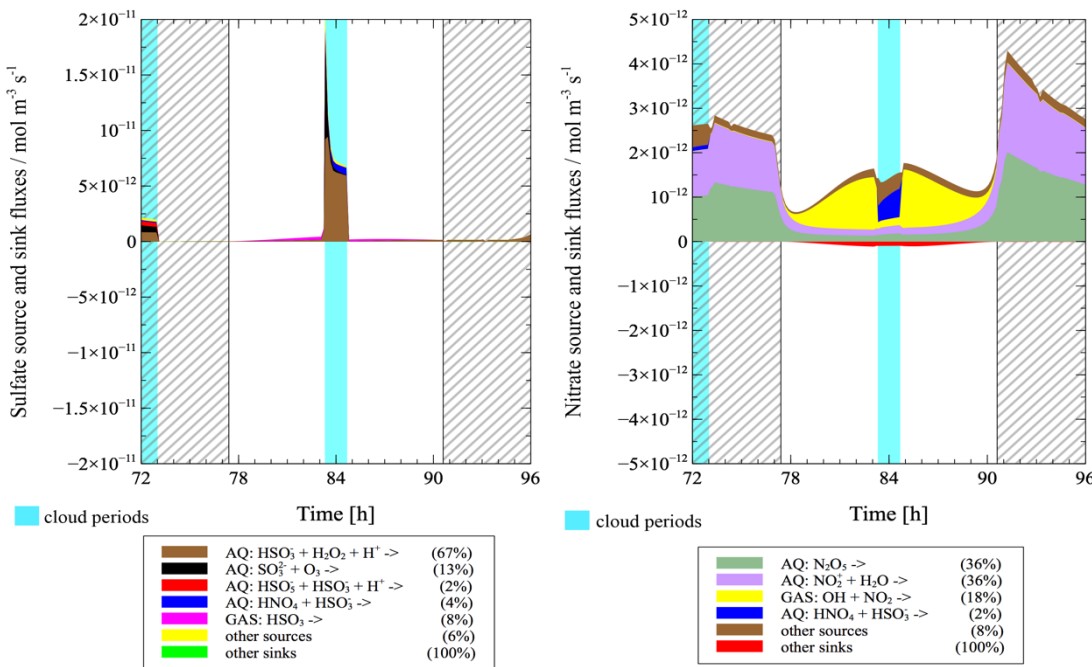

**Figure 4. Modeled multiphase (gas + aqueous phase) source and sink fluxes of sulfate and nitrate (light blue column: cloud; shadow: night; percent is for the third model day).**





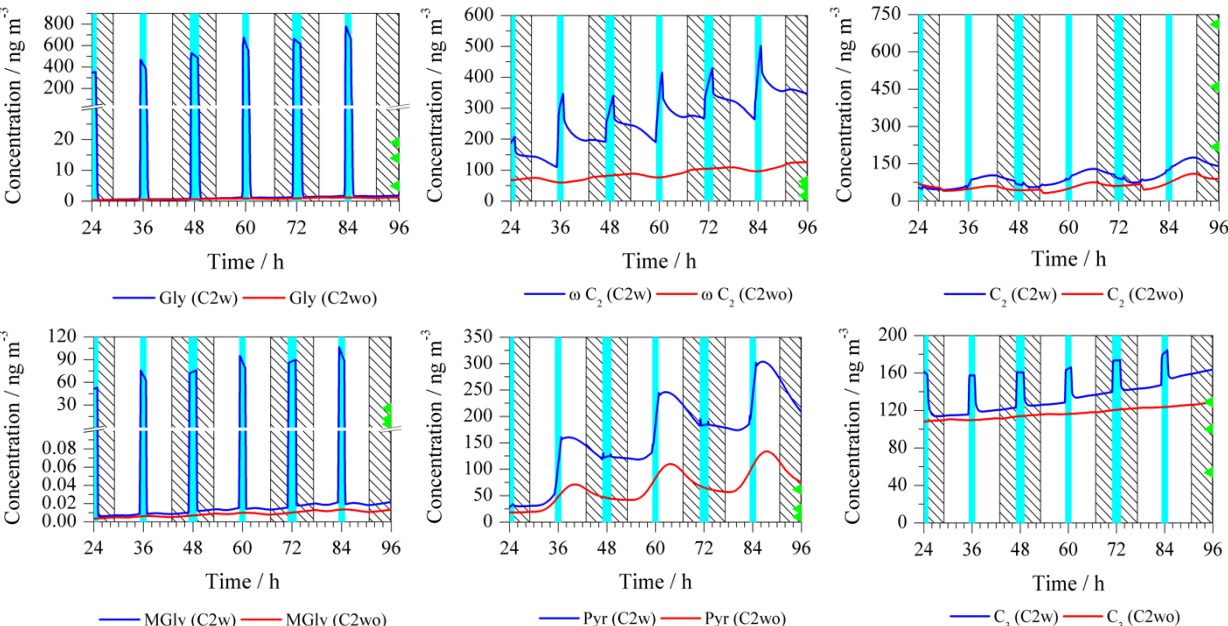

**Figure 5. Time series of the modeled concentrations of selected DCRCs (top: Gly, ωC₂, C₂; bottom: MGly, Pyr and C₃) in the C2w and C2wo cases (light blue column: cloud; shadow: night; green triangle: the maximum (above), average (middle) and minimum (below) value of measured concentrations at Mt. Tai).**






**Figure 6. Modeled multiphase (gas + aqueous phase) source and sink fluxes of ωC₂ (above left), C₂ (above right), Pyr (below left) and C₃ (below right) along the trajectory of the third model day (light blue column: cloud; shadow: night; percentage is for the third model day).**








**Figure 7. Concentration variations of modeled sulfate, nitrate, ammonium, Gly, ωC₂, C₂, MGly, Pyr and C₃ when doubling emission data (light blue column: cloud; shadow: night).**






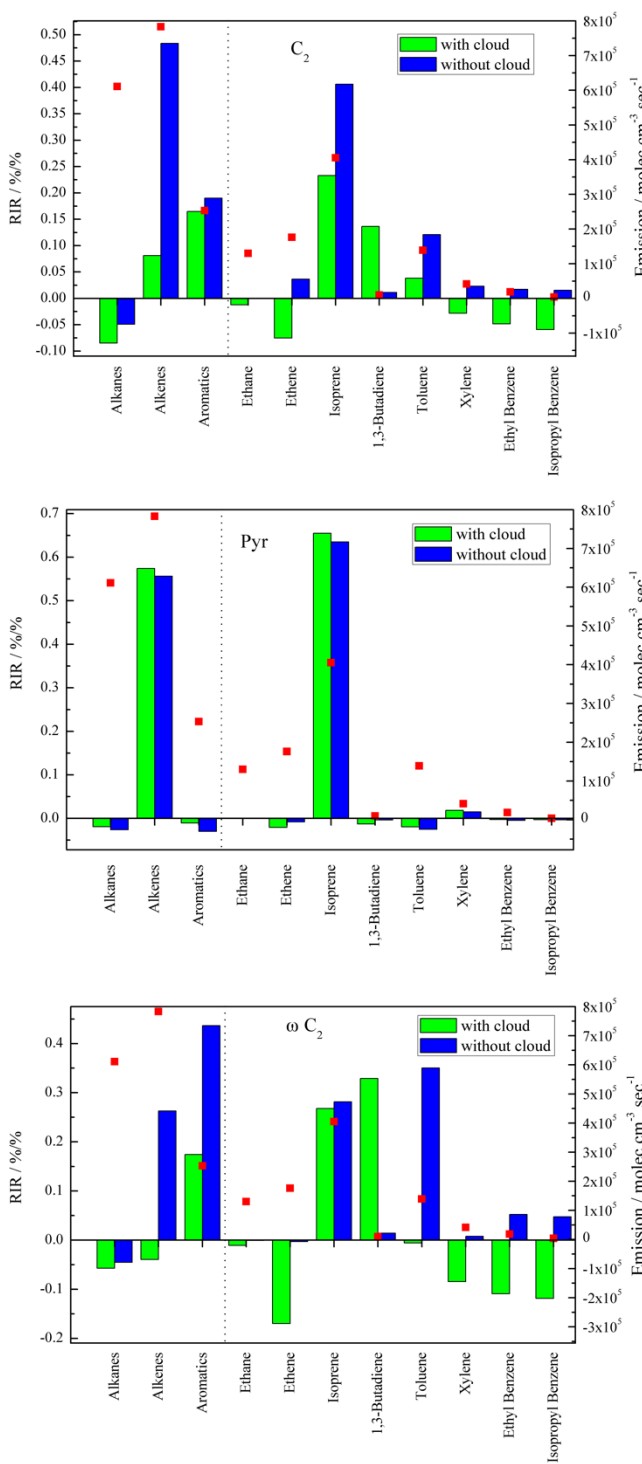

**Figure 8. The calculated RIRs for C$_2$, Pyr and ωC$_2$ in both the C2w (green bars) and C2wo (blue bars) cases at Mt. Tai (column: RIR values; red dots: emission data).**






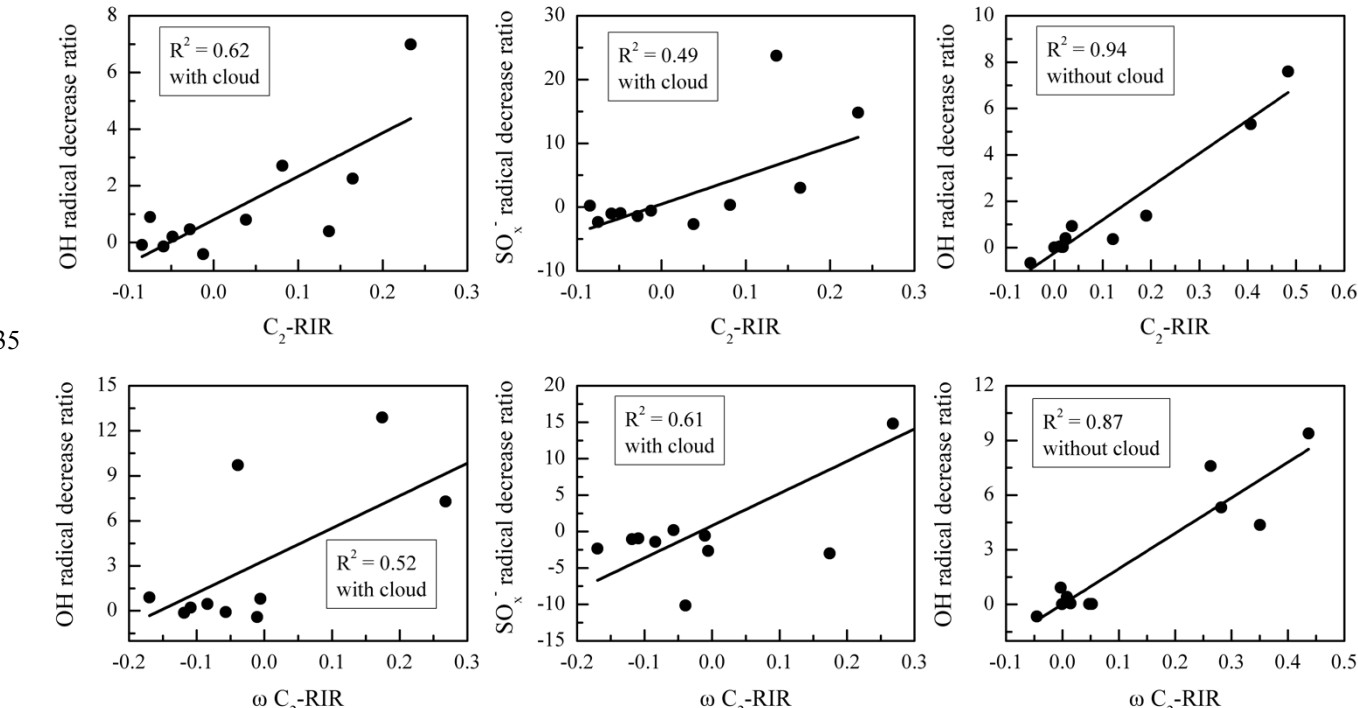

**Figure 9. Correlations between the decreasing ratios of radical oxidants and C$_2$-RIR (above) and ωC$_2$-RIR (below) under C2w and C2wo conditions, respectively.**





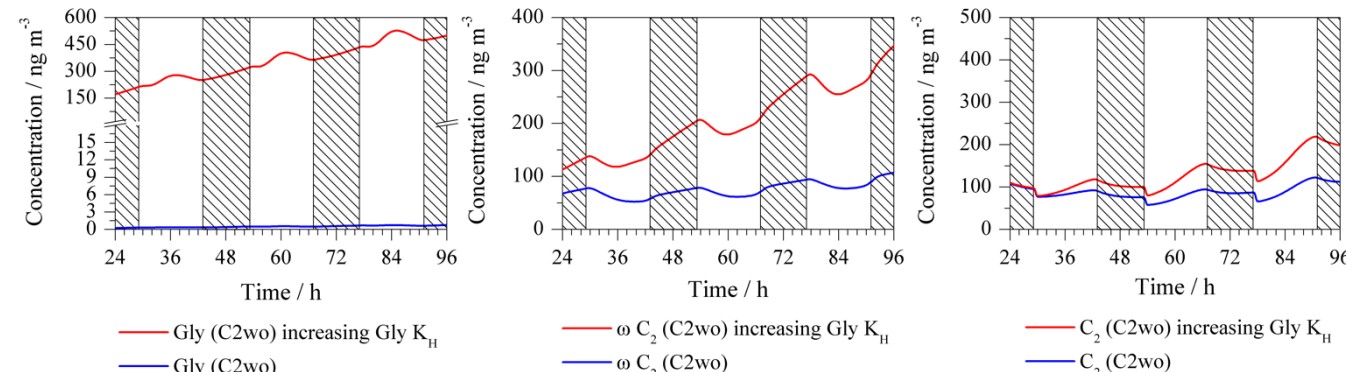


**Figure 10. Concentration variations of modeled Gly, ωC₂ and C₂ when increasing Gly' Henry law constant by two orders of magnitude (shadow: night; K_H: Henry law constant).**