# Peer review of "Multiphase MCM/CAPRAM modeling of formation and processing of secondary aerosol constituents observed at the Mt. Tai summer campaign 2014"

_Atmospheric Chemistry and Physics, 2019_

## Referee Comment (RC2)

**Review for paper:**
*"Multiphase MCM/CAPRAM modeling of formation and processing of secondary aerosol constituents observed at the Mt. Tai summer campaign 2014"*
*by*
*Y. Zhu et al.*

*Submitted to Atmospheric Chemistry and Physics*

**General Comments:**
This paper presents a modeling study with a 0D-sophisticated model of the formation and processing of dicarboxylic acids and related compounds (oxo-carboxylic acids and α-dicarbonyls) (DCRCs) in clouds and deliquesced aerosols at the Mt. Tai. Results are compared to observations of aerosols composition analyzed from night and day filters. This study is very interesting and valuable in doing the effort to compare simulated results to observations. However, I have some main concerns detailed below on the definition of the scenario and on the absence of information on the microphysical structure of simulated cloud.

**Overall recommendation:**
I recommend that the paper should be accepted for publication in Atmospheric Chemistry and Physics after major and specific revisions listed below.

**Major Comments:**
1 – About the simulations set-up:
I found that the conditions used in simulations (meteorological conditions, emissions, initial chemical concentrations, aerosol parameters) are very confused. I list below some of my main concerns:

- No value is given for meteorological conditions contrary to that it states page 5, line 148. I did not understand why the meteorological values along the trajectories are not directly used to drive the air parcel. For instance, in Zhu et al. (2018), WRF simulations and HYSPLIT back-trajectories are mentioned: I guess that typical trajectories to Mt. Tai could be extracted from these runs.
- With regard to scenarios (cloud or cloud-free), in my opinion, a third series of simulation without aqueous chemistry at all should be of interest in assessing the contribution of aqueous pathways to the formation of secondary aerosol constituents. Especially since the ideal solution hypothesis biases the results for aqueous chemistry inside deliquesced aerosols.
- For biogenic emissions, you should precise, which inventory you used to extract biogenic emissions: is-it MEGAN-MACC, CAMS-GLOB-BIO? Also, why did you consider biogenic emissions only for isoprene and pinenes whereas a lot of other species are available from MEGAN-MACC or CAMS-GLOB-BIO? I suggest that the biogenic and anthropogenic emission values be separated in Table S1.
- Where values for deposition velocities come from?
- In Table S3 and S4, the source of each value (literature or urban CAPRAM scenario) should be indicated. Why measured observations of VOC analysed from stainless steel canisters (Zhu et al., 2018) are not used to initialize the model?
- I do not understand aerosol parameters in Table S4. Although, the unit of ng/m3 or μm/m3 is used in the article, aerosol species composition is given in mixing ratio (g/g) in the Table. No initial value is given for aerosol number concentrations.

For all these reasons, I found the simulation set-up questionable.

2 – About the results:

In my opinion, some important information is missing. No results are reported for the microphysical structure of simulated clouds and how they compare to observations.

**Specific Revisions:**
**Introduction:**

- The recent following review paper should be cited: Shrivastava, M., Cappa, C. D., Fan, J., Goldstein, A. H., Guenther, A. B., Jimenez, J. L., Kuang, C., Laskin, A., Martin, S. T., Ng, N. L., Petaja, T., Pierce, J. R., Rasch, P. J., Roldin, P., Seinfeld, J. H., Shilling, J., Smith, J. N., Thornton, J. A., Volkamer, R., Wang, J., Worsnop, D. R., Zaveri, R. A., Zelenyuk, A. and Zhang, Q.: Recent advances in understanding secondary organic aerosol: Implications for global climate forcing, Rev. Geophys., 2016RG000540, doi:10.1002/2016RG000540, 2017.
- The citation of following paper should be added in the paragraph lines 52 to 64: Mouchel-Vallon, C., Deguillaume, L., Monod, A., Perroux, H., Rose, C., Ghigo, G., Long, Y., Leriche, M., Aumont, B., Patryl, L., Armand, P. and Chaumerliac, N.: CLEPS 1.0: A new protocol for cloud aqueous phase oxidation of VOC mechanisms, Geosci. Model Dev., 10(3), 1339–1362, doi:10.5194/gmd-10-1339-2017, 2017.

**Part 2:**

- The sampling period of observations should be given here. A discussion about the known sources of DCRCs deduced from Zhu et al. (2018) should be done.
- 2.1:
  - The limitation of SPACCIM due to the assumptions of ideal solutions concerns deliquesced particles. This should be specified.
  - Microphysical processes include in SPACCIM should be listed. Also it should be mentioned that cloud particles size distribution is spectral.
  - Does SPACCIM include aerosols? If yes, it should be specified how: which processes are considered for aerosols: microphysics processes (nucleation, aggregation, sedimentation), chemical aging, nucleation and impaction scavenging by cloud particles? Also the method to represent their size distribution should be indicated. Do you use thermodynamics equilibrium to partition inorganic and organic species between gas phase and particles?
  - Could you please indicate if recent findings on isoprene and aromatics gas phase chemistry are included in MCM? As the reference papers for MCM are from 2003, it is probably not the case and could lead to other limitations in the results.
- 2.2: Could you please indicate the reason to exclude days of the campaign influenced by biomass burning?
- 2.3: You should use the proper ECCAD address; https://eccad.aeris-data.fr/. Note that you have to acknowledge ECCAD and to reference the dataset citation (see metadata page of the inventory that you used). Why did you use urban CAPRAM scenario for missing values? I suggest to indicate also initial gas-phase conditions in mixing ratio (ppbv or pptv), which is the usual quantity used for representing observations of trace gases.

**Part 3:**

- 3.1.1:
  - Page 6, lines 178-179, it is stated: "The reduction of OH radical is mainly caused by the reduction of the gas-phase formation pathway of the $HO_2$+ NO

reaction" and page 7, lines 198-200: "Under daytime cloud droplet conditions, OH aqueous-phase concentrations are increased by a factor of 3, mainly due to the increased direct transfer of OH from the gas phase.". I found these statements contradictory. I guess that a plot showing the total OH concentrations (gaseous + aqueous) would clarify this point.

- o Whereas, the text page 6 line 187 indicates that obtained OH and HO2 gaseous concentrations are compared to available measurements, it is mainly modelling studies that are used for this comparison. Moreover, no details are given on these studies: which model, which conditions (period of simulation, chemical mechanism used for instance). The only observations cited show discrepancies with results, in particular for OH.

- o Whereas some estimations exist of OH concentration in cloud droplets (see Arakaki, T., Anastasio, C., Kuroki, Y., Nakajima, H., Okada, K., Kotani, Y., Handa, D., Azechi, S., Kimura, T., Tsuhako, A. and Miyagi, Y.: A General Scavenging Rate Constant for Reaction of Hydroxyl Radical with Organic Carbon in Atmospheric Waters, Environ. Sci. Technol., 47(15), 8196–8203, doi:10.1021/es401927b, 2013), no comparison is discussed. It seems to me that the simulated OH concentrations in droplets are high in comparison to available estimations.

- 3.1.2: I suggest using mixing ratio for gas-phase instead of concentrations when comparing simulated results with in-situ measurements. As mixing ratio is a relative quantity, it is not dependent on altitude (via pressure and temperature).

- 3.2: As no information is given on how the interactions between aerosols and cloud water are considered, it is difficult to interpret the results. Why on Fig. 3 sulphate remains constant when cloud dissipates whereas nitrate decreases at the same time?

- 3.3.1: I suggest recalling in legend of Fig.5 what Gly, wC2, C2, Pyr, MGly and C3 means. Does this figure show the aerosol mass concentrations as shown in Figure 3? If yes, I do not understand why the text page 11, line 316 refers to aqueous phase concentrations. Does this mean that aqueous phase concentration is the same than aerosol mass concentration? These results are difficult to interpret without knowing if thermodynamics equilibrium between gas phase and aerosols is considered for organics species. For instance, for malonic acid, it should be mainly in the aerosols and not in the gas phase (Limbeck, A., Kraxner, Y. and Puxbaum, H.: Gas to particle distribution of low molecular weight dicarboxylic acids at two different sites in central Europe (Austria), Journal of Aerosol Science, 36, 991-1005, 2005).

  - o Dicarbonyl compounds: I find that the remarkable result is that Gly and MGly are very similar concentrations at the end of the simulation considering or not cloud chemistry. In addition, the small increase of MGly in the cloud case in comparison to the no cloud one, seems to be related to its small production inside cloud droplets during night. So, I disagree with this statement: "This might have been caused by the fact that the aqueous oxidation fluxes under nighttime cloud conditions are lower than the ones under daytime".

  - o C2 carboxylic acids: Could you please rewrite this passage, which is not clear, only describes the curves and does not give hypotheses about the observed trends.

  - o C3 carboxylic acids: I don't see an increase of Pyr in the nighttime cloud.

- 3.3.2:

  - o As described in Zhu et al. (2018), DCRCs observations are available for daytime and nighttime. Thus, I suggest indicating in Table 3, ratio for day and

night. Could you please specify on which simulated period the averages are done (the three days or only the third day)?

- By this sentence "SPACCIM overestimates the measured ωC2 concentrations, but underestimates the measured C2 ones, suggesting the conversion of ωC2 might be implemented less efficiently into CAPRAM.", do you mean that the aqueous phase oxidation of ωC2 producing C2 implemented in CAPRAM seems to be less efficient than in the field?

- Page 12, lines 356-362: the hypothesis of missing processes enhancing the partitioning of MGly implies that the model is capable of simulating realistic total MGly concentrations (gas phase + aerosols): is-it the case?

- I disagree with this statement "The emission data are obtained through model calculations, not field measurements". Emission data comes from inventory, which is developed in part based upon measurements. I guess that the more probable error coming from emission data is due to the horizontal and temporal resolution of inventories used. This point should be discussed.

- Why do you mean by "The height of Mt. Tai (about 1500 m) also causes its input to be indefinite."?

- MCM mechanism: I guess you could cite other limitations, in particular in biogenic VOC oxidation as the Mt. Tai is surrounded by deciduous forest.

- I disagree that ratios are acceptable except for C3. I think that it is an interesting result to show that, even a sophisticated model as the one used in this study, is not able to reproduce DCRCs observations. A discussion trying to assess why the C3 ratio is close to 1 should be interesting.

- 3.4.1:
  - ωC2: I see a low net formation flux during the last night for non-cloud period.
  - C2: Please moderate the last sentence: one of the reasons and not the reason.
  - Pyr: Do you consider the photolysis of Pyr in aqueous phase (Reed Harris, A. E., Ervens, B., Shoemaker, R. K., Kroll, J. A., Rapf, R. J., Griffith, E. C., Monod, A. and Vaida, V.: Photochemical Kinetics of Pyruvic Acid in Aqueous Solution, J. Phys. Chem. A, doi:10.1021/jp502186q, 2014.) ?

- 3.5.1: Green lines on Fig. 7 are difficult to see: could you please use another colour (grey for instance)?

- 3.5.2: Could you please explain how negative and positive RIR values have to be interpreted? It would help the reader to follow the discussion on DCRCs precursors. Page 17, line 519: please suppress As.

---

## Referee Comment (RC1) · Anonymous Referee #1 · 30 Dec 2019

Review of Zhu et al. "Multiphase MCM/CAPRAM modeling of formation and processing of secondary aerosol constituents observed at the Mt. Tai summer campaign 2014"

The authors report a detailed multiphase chemical modeling study of the formation and processing of secondary aerosol compositions during transport to the Mt. Tai in summer 2014. The model performance of MCM/CAPRAM was evaluated against the field observations, and the day vs. night and with cloud vs. non-cloud processes were examined. The major formation pathways and key precursors of sulfate, nitrate, ammonium, and DCRCs were identified with the model. The impacts of emissions and

glyoxal partitioning constants on the modeling results were also estimated by sensitivity studies. Despite an increasing number of field observational studies of secondary aerosols in recent years in China, such kind of detailed multiphase modeling study is still lacking. This study is helpful for better understanding the regional formation and processing of secondary inorganic and organic aerosols in the North China Plain. Therefore I would recommend that this manuscript can be considered for publication at ACP after the following specific comments being addressed.

Specific Comments:

P2 L58 "However, formation pathways based on measured data...": rephrase this sentence.

P3 L67-68: this sentence is not clear. Do the Yangtze River Delta and Bohai Rim have a total of 410 million populations? Additionally, the commonly used three largest economic zones in China don't include the Bohai Rim.

P3 L83-84: provide the standard deviations for the average temperature and RH values.

P4 L98: predicted and observed concentrations...

P4 Section 2.2: I suggest the authors to provide the air mass cluster figures in the SI so that the readers can easily access the plot.

P4 L111-112: was only an important source...

P5 L150-156: this paragraph is a little bit redundant with the last paragraph of the Introduction (P3 L73-79). I suggest the authors may remove this paragraph.

P6 L158: replace "oxidations" by "oxidant"

P6 L 168: non-radical oxidant concentrations...

P6 L 175: I would suggest the authors to delete the citation here as it is only modeling

results from this study.

Section 3.2: I suggest to provide the sub-titles for "Sulfate" (e.g., 3.2.1), "Nitrate" and "Ammonium".

P9 L253: replace "dominated" by "dominant"

P10, L286-287 "Potential reasons are discussed below": it is not clear where the potential reasons are discussed. Please clarify.

P11 L 321: replace "shows" by "show"

P11 L348-349: I suggest to move this sentence to the beginning of this section, i.e., L319.

P12 L363: under-estimation

P12 L365-366: I presume the emission data were obtained from the emission inventory, rather than model calculations.

P13 L401 "reported in in above references": delete one "in".

---

## Author Comment (AC1) · 28 Feb 2020

**Responses to the reviewer comments on**

**"Multiphase MCM/CAPRAM modeling of formation and processing of secondary aerosol constituents observed at the Mt. Tai summer campaign 2014" by Zhu et al.**

The authors would like to thank both reviewers for their constructive and good suggestions to improve our manuscript. We have carefully considered all the review comments and revised the manuscript. Below, we provide responses to the comments in blue, with changes made in the manuscript highlighted in red.

**Response to Reviewer 1:**

*The authors report a detailed multiphase chemical modeling study of the formation and processing of secondary aerosol compositions during transport to the Mt. Tai in summer 2014. The model performance of MCM/CAPRAM was evaluated against the field observations, and the day vs. night and with cloud vs. non-cloud processes were examined. The major formation pathways and key precursors of sulfate, nitrate, ammonium, and DCRCs were identified with the model. The impacts of emissions and glyoxal partitioning constants on the modeling results were also estimated by sensitivity studies. Despite an increasing number of field observational studies of secondary aerosols in recent years in China, such kind of detailed multiphase modeling study is still lacking. This study is helpful for better understanding the regional formation and processing of secondary inorganic and organic aerosols in the North China Plain. Therefore I would recommend that this manuscript can be considered for publication at ACP after the following specific comments being addressed.*

Response: We thank reviewer#1 for the helpful comments. Below, we address the comments and have revised the manuscript accordingly. For clarity, the reviewer's comments are listed below in *black italics*, whereas our responses and changes in manuscript are shown in blue and red, respectively. Revised Tables and Figures are put in the end.

*1. P2 L58 "However, formation pathways based on measured data are still limited": rephrase this sentence.*

Response: We have rephrased this sentence as follows:

However, modeling studies that focus on understanding of DCRCs formation pathways based on field measurements are limited.

(Page 2, Line 58-59)

*2. P3 L67-68: this sentence is not clear. Do the Yangtze River Delta and Bohai Rim have a total of 410 million populations? Additionally, the commonly used three largest economic zones in China don't include the Bohai Rim.*

Response: Yes, the intended meaning was to say that both together have a population of around 410 million. In addition, we checked the China Statistical Yearbook in 2019, and found that together, the Yangtze River Delta and Bohai Rim regions had a population of more than 450 million in 2018. Therefore, we changed the sentence in the revised manuscript as follows:

Mt. Tai is located in Shandong province in the NCP, and between the Bohai Rim (BHR) and the Yangtze River Delta (YRD) regions. Together, the BHR and YRD regions had a population of more than 450 million in 2018 (China Statistical Yearbook in 2019).

(Page 3, Line 67-71)

*3. P3 L83-84: provide the standard deviations for the average temperature and RH values.*

Response: We have added standard deviations for the average temperature and RH values as follows:

$17 \pm 6.2 \, °C$

(Page 3, Line 87)

$87 \pm 13 \, \%$

(Page 3, Line 88)

*4. P4 L98: predicted and observed concentrations......*

Response: We have changed "concentration" to "concentrations" in the revised manuscript as follows:

These limitations have to be kept in mind when studying deliquesced particles and comparing predicted and observed concentrations at Mt. Tai.

(Page 4, Line 113-114)

*5. P4 Section 2.2: I suggest the authors to provide the air mass cluster figures in the SI so that the readers can easily access the plot.*

Response: We have added air mass cluster Figure in the supplement as follows:

[Figure]

Figure S1. Three-day back-trajectories for Mt. Tai during the sampling period (green triangle: Mt. Tai).

*6. P4 L111-112: was only an important source.....*

Response: We have added "an" in the revised manuscript as follows:

Additionally, Zhu et al. (2018) have clearly shown that biomass burning was only an important source during the first half of the sampling period (June 4 – 19).

(Page 4, Line 128, Page 5, Line 129)

*7. P5 L150-156: this paragraph is a little bit redundant with the last paragraph of the Introduction (P3 L73-79). I suggest the authors may remove this paragraph.*

Response: According to the reviewer suggestion, we have removed this paragraph.

(Page 6, Line 179-185)

*8. P6 L158: replace "oxidations" by "oxidant"*

Response: We have replaced "oxidations" by "oxidant" in the revised manuscript as follows:

Due to the key role of radical and non-radical oxidant in the formation processes of secondary aerosol constituents, their concentration variations and corresponding reasons are investigated.

(Page 6, Line 187-188)

*9. P6 L 168: non-radical oxidant concentrations…*

Response: We have added "oxidant" in the revised manuscript as follows:

To our knowledge, this is the first detailed multiphase chemical modeling study examining radical and non-radical oxidant concentrations along the trajectory to the Mt. Tai under day vs. night and cloud vs. non-cloud cases.

(Page 7, Line 197-198)

*10. P6 L 175: I would suggest the authors to delete the citation here as it is only modeling results from this study.*

Response: The citations have been deleted.

(Page 7, Line 204-206)

*Section 3.2: I suggest to provide the sub-titles for "Sulfate" (e.g., 3.2.1), "Nitrate" and "Ammonium".*

Response: We have added sub-titles for "Sulfate", "Nitrate" and "Ammonium" subsections as follows:

3.2.1 Sulfate

(Page 9, Line 282)

3.2.2 Nitrate

(Page 10, Line 305)

3.2.3 Ammonium

(Page 11, Line 331)

*11. P9 L253: replace "dominated" by "dominant"*

Response: We have replaced "dominated" by "dominant" in the revised manuscript as follows:

In the nighttime cloud, aqueous-phase reaction of $HSO_3^-$ with $H_2O_2$ (42 %), and aqueous reaction of bisulfite with $O_3$ (28 %) are dominant pathways for sulfate formation.

(Page 10, Line 295-297)

*12. P10, L286-287 "Potential reasons are discussed below": it is not clear where the potential reasons are discussed. Please clarify.*

Response: We have changed the sentence in the revised manuscript as follows:

Potential reasons are discussed in Sect. 3.3.2.

(Page 11, Line 329-330)

*13. P11 L 321: replace "shows" by "show"*

Response: We have replaced "shows" by "show" in the revised manuscript as follows:

In the C2w case, Gly and MGly concentration patterns show a substantial uptake into cloud droplets.

(Page 12, Line 366)

*14. P11 L348-349: I suggest to move this sentence to the beginning of this section, i.e., L319.*

Response: According to the reviewer suggestion, we have moved the sentence.

(Page 12, Line 362-364)

*15. P12 L363: under-estimation*

Response: We have changed "underestimation" to "under-estimation" in the revised manuscript as follows:

The over- and under-estimation of the measured concentrations of inorganic and organic aerosol constituents could have the following reasons.

(Page 13, Line 414-415)

*16. P12 L365-366: I presume the emission data were obtained from the emission inventory, rather than model calculations.*

Response: We have changed "model calculations" to "emission inventory" in the revised manuscript as follows:

The emission data are obtained through a new anthropogenic emission inventory in Asia, which provides monthly emissions in 2010 by sector at 0.25° × 0.25° resolution.

(Page 13, Line 416-418)

*17. P13 L401 "reported in in above references": delete one "in".*

Response: deleted.

(Page 15, Line 456)

---

## Author Comment (AC2) · 28 Feb 2020

**Responses to the reviewer comments on**

**"Multiphase MCM/CAPRAM modeling of formation and processing of secondary aerosol constituents observed at the Mt. Tai summer campaign 2014" by Zhu et al.**

The authors would like to thank both reviewers for their constructive and good suggestions to improve our manuscript. We have carefully considered all the review comments and revised the manuscript. Below, we provide responses to the comments in blue, with changes made in the manuscript highlighted in red.

**Response to Reviewer 2:**

*This paper presents a modeling study with a 0D-sophisticated model of the formation and processing of dicarboxylic acids and related compounds (oxo-carboxylic acids and α-dicarbonyls) (DCRCs) in clouds and deliquesced aerosols at the Mt. Tai. Results are compared to observations of aerosols composition analyzed from night and day filters. This study is very interesting and valuable in doing the effort to compare simulated results to observations. However, I have some main concerns detailed below on the definition of the scenario and on the absence of information on the microphysical structure of simulated cloud. I recommend that the paper should be accepted for publication in Atmospheric Chemistry and Physics after major and specific revisions listed below.*

Response: We appreciate the reviewer for the helpful comments and suggestions. Below we address the comments and have revised the manuscript accordingly. For clarity, the reviewer's comments are listed below in *black italics*, while our responses and changes in manuscript are shown in blue and red, respectively. Revised Tables and Figures are in the end.

*1. 1 – About the simulations set-up: I found that the conditions used in simulations (meteorological conditions, emissions, initial chemical concentrations, aerosol parameters) are very confused. I list below some of my main concerns: No value is*

*given for meteorological conditions contrary to that it states page 5, line 148. I did not understand why the meteorological values along the trajectories are not directly used to drive the air parcel. For instance, in Zhu et al. (2018), WRF simulations and HYSPLIT back-trajectories are mentioned: I guess that typical trajectories to Mt. Tai could be extracted from these runs.*

Response: We have used the HYSPLIT back-trajectories results (see Fig. S1 in the supplement) given in Zhu et al. (2018). According to HYSPLIT back-trajectories results in Zhu et al. (2018), we selected clusters 2 and 4 to simulate and investigate the formation processes and the fate of DCRCs. Because the two clusters accounted for 79 % of the total trajectories. Moreover, the sum of DCRC concentrations in clusters 2 and 4 amounted to 73 % of total DCRC concentration during the sampling period (Zhu et al., 2018).

The meteorological data along the cluster trajectories derived by HYSPLIT are mean values of many trajectories. The resolution of HYSPLIT is 1°x1° and the DCRC measurements at Mt. Tai have a 12 h resolution only. Due to the coarse spatial- and time-resolution of both data, we decided to use a representative trajectory for each cluster instead of single trajectories.

Cloud conditions are assumed from satellite pictures, because HYSPLIT data have a very coarse spatial resolution which does not allow the identification of cloud conditions along the trajectory. The relative humidity in the 1°x1° grid cells represents a mean value. Therefore, a grid cell with sub-grid clouds would finally also be characterized by RH values below 100% as only a certain fraction is filled with clouds. In order to illustrate the applied meteorological data in each scenario, we have added the following Figure S2 in the supplement.

[Figure]

Figure S2. Meteorological data in different scenarios.

We also changed corresponding sentence as follows:

These initial model data, and also aerosol parameters are given in Table S3 and Table S4. The meteorological scenarios are illustrated in Figure S2.

(Page 6, Line 176-177)

Reference:

Zhu, Y., Yang, L., Chen, J., Kawamura, K., Sato, M., Tilgner, A., van Pinxteren, D., Chen, Y., Xue, L., Wang, X., Simpson I. J., Herrmann, H., Blake D. R., and Wang, W. X.: Molecular distributions of dicarboxylic acids, oxocarboxylic acids and α-dicarbonyls in PM2.5 collected at the top of Mt. Tai, North China, during the wheat burning season of 2014, Atmos. Chem. Phys., 18, 10741-10758, 2018.

*2. 1 – About the simulations set-up: With regard to scenarios (cloud or cloud-free), in my opinion, a third series of simulation without aqueous chemistry at all should be of interest in assessing the contribution of aqueous pathways to the formation of secondary aerosol constituents. Especially since the ideal solution hypothesis biases the results for aqueous chemistry inside deliquesced aerosols.*

Response: The applied gas-phase mechanism MCM is a mechanism that is not able to produce dicarboxylic acids, because of the exclusive treatment of organic acid oxidation via H-abstraction from the OH group destroying the carboxylic acid functionality. As a consequence, only monocarboxylic acids can be formed by gas-phase chemistry within the applied multiphase chemistry system. Therefore, it is evident that most of the compounds in this study are mainly or, in case of the DCRCs, exclusively formed in the aqueous-phase. Thus, in author's opinions, a simulation without aqueous chemistry is not useful. Thus, a comparison with aqueous-phase simulations might result into misleading interpretations.

*3. 1 – About the simulations set-up: For biogenic emissions, you should precise, which inventory you used to extract biogenic emissions: is-it MEGAN-MACC, CAMS-GLOB-BIO? Also, why did you consider biogenic emissions only for isoprene and pinenes whereas a lot of other species are available from MEGAN-MACC or CAMS-GLOB-BIO? I suggest that the biogenic and anthropogenic emission values be separated in Table S1.*

Response: (1) Biogenic emission data are obtained from MEGAN-MACC, we have

changed the corresponding sentence in the revised manuscript as follows:

Biogenic emission data (isoprene, α- and β-pinenes) are obtained from Emissions of atmospheric Compounds and Compilation of Ancillary Data (ECCAD), MEGAN-MACC dataset (https://eccad.aeris-data.fr/).

(Page 5, Line 158-160)

(2) MEGAN-MACC includes 21 biogenic species, such as ethane, propane, propene, toluene. However, most of the treated compounds are mainly from anthropogenic sources, especially in China. For example, propane is the main components of liquefied petroleum gas/natural gas (McCarthy et al., 2013), ethane, propene and toluene originate from automobile exhaust (Chang et al., 2009), ketones and aldehydes are produced from anthropogenic secondary oxidation (Liu et al., 2015). Therefore, emission data of these species are obtained from a new anthropogenic emission inventory in Asia (Li et al., 2017), not from MEGAN-MACC.

Isoprene and pinenes are mainly from biogenic sources (Brown et al., 2007; Zhang et al., 2013). The new anthropogenic emission inventory in Asia focuses on anthropogenic sources, which inputs to be indefinite in biogenic emission (Li et al., 2017). Therefore, emission of isoprene and pinenes are obtained from MEGAN-MACC.

References:

Brown, S. G., Frankel, A., Hafner, H. R., 2007. Source apportionment of VOCs in the Los Angeles area using positive matrix factorization. Atmospheric Environment 41, 227-237, 2007.

Chang, C. C., Wang, J. L., Lung, S. C. C., Liu, S. C., Shiu, C. J.: Source characterization of ozone precursors by complementary approaches of vehicular indicator and principal component analysis. Atmospheric Environment 43 (10), 1771-1778, 2009.

Li, M., Zhang, Q., Kurokawa, J., Woo, J., He, K. B., Lu, Z. F., et al.: MIX: a mosaic Asian anthropogenic emission inventory under the international collaboration framework of the MICS-Asia and HTAP. Atmospheric Chemistry and Physics, 17, 935–963, 2017.

Liu, Y., Yuan, B., Li, X., Shao, M., Lu, S., Li, Y., et al.: Impact of pollution controls in

Beijing on atmospheric oxygenated volatile organic compounds (OVOCs) during the 2008 Olympic Games: observation and modeling implications. Atmospheric Chemistry and Physics, 15, 3045–3062, 2015.

McCarthy, M. C., Aklilu, Y. A., Brown, S. G., Lyder, D. A.: Source apportionment of volatile organic compounds measured in Edmonton, Alberta. Atmospheric Environment 81 (2), 504-516, 2013.

Zhang, J. K., Sun, Y., Wu, F. K., Sun, J., Wang, Y. S.: The characteristics, seasonal variation and source apportionment of VOCs at Gongga Mountain, China. Atmospheric Environment, 1-9, 2013.

(3) We have separated biogenic and anthropogenic emission values in revised Table S1 as follows:

Table S1. Emission data applied in the SPACCIM ([#]: anthropogenic emission values, [*]: biogenic emission values).

| Compound | Emission / molec cm$^{-3}$ s$^{-1}$ | Compound | Emission / molec cm$^{-3}$ s$^{-1}$ |
|---|---|---|---|
| Acetone[#] | 3.51E+04 | Acetaldehyde[#] | 1.44E+04 |
| Ethane[#] | 1.30E+05 | Ethylene[#] | 1.76E+05 |
| Propane[#] | 1.87E+05 | Glyoxal[#] | 1.04E+04 |
| n-Butane[#] | 6.79E+04 | Formaldehyde[#] | 3.15E+04 |
| Isobutane[#] | 2.99E+04 | Biacetyl[#] | 1.49E+03 |
| 2,2-Dimethyl Butane[#] | 2.00E+03 | Benzaldehyde[#] | 3.37E+02 |
| Isopentane[#] | 6.75E+04 | Methacrolein[#] | 2.08E+03 |
| n-Pentane[#] | 2.67E+04 | Methyl ethyl ketone[#] | 8.68E+03 |
| 2-Methyl Pentane[#] | 1.57E+04 | Methanol[#] | 2.28E+04 |
| 3-Methylpentane[#] | 1.10E+04 | Methylglyoxal[#] | 3.93E+03 |
| n-Hexane[#] | 6.28E+03 | Methyl Vinyl Ketone[#] | 2.62E+02 |
| n-Heptane[#] | 4.71E+03 | Propene[#] | 2.96E+04 |
| 2,3-Dimethyl Butane[#] | 4.71E+03 | 1-Hexene[#] | 2.45E+04 |
| n-Decane[#] | 1.77E+04 | 1-Butene[#] | 1.22E+04 |
| 3-Methyl Hexane[#] | 1.77E+04 | 1-Pentene[#] | 1.12E+04 |
| n-Nonane[#] | 6.45E+03 | 3-Methyl-1-Butene[#] | 3.06E+03 |
| n-Octane[#] | 6.45E+03 | cis-2-Pentene[#] | 2.25E+04 |
| 2-Methyl Hexane[#] | 4.84E+03 | trans-2-Pentene[#] | 2.25E+04 |
| n-Dodecane[#] | 3.22E+03 | 1,3-Butadiene[#] | 9.64E+03 |
| n-Undecane[#] | 1.61E+03 | 2-Methyl-2-Butene[#] | 8.03E+03 |
| Toluene[#] | 1.39E+05 | Cis-2-Hexene[#] | 8.03E+03 |
| Ethyl Benzene[#] | 1.86E+04 | Trans-2-Hexene[#] | 8.03E+03 |

| | | | |
|---|---|---|---|
| n-Propyl Benzene[#] | 7.43E+03 | Propionaldehyde[#] | 6.40E+03 |
| Isopropyl Benzene[#] | 3.72E+03 | Limonene[#] | 3.28E+02 |
| m-Xylene[#] | 1.46E+04 | Carbon monoxide[#] | 3.04E+07 |
| p-Xylene[#] | 1.46E+04 | Carbon dioxide[#] | 1.15E+09 |
| o-Xylene[#] | 1.23E+04 | Ammonia[#] | 3.81E+06 |
| 1,2,3-Trimethyl Benzene[#] | 1.01E+04 | Nitric Oxide[#] | 2.51E+05 |
| 1,3,5-Trimethyl Benzene[#] | 1.01E+04 | Nitrogen dioxide[#] | 1.42E+06 |
| m-Ethyl Toluene[#] | 5.61E+03 | Sulfur dioxide[#] | 1.91E+06 |
| o-Ethyl Toluene[#] | 5.61E+03 | Isoprene[*] | 4.05E+05 |
| p-Ethyl Toluene[#] | 5.61E+03 | a-pinene[*] | 2.99E+04 |
| 1,2,4-Trimethyl Benzene[#] | 5.61E+03 | β-pinene[*] | 1.28E+04 |

*4. 1 – About the simulations set-up: Where values for deposition velocities come from?*

Response: Dry deposition velocities were taken from Ganzeveld et al. (1998). We have changed the corresponding sentence as follows:

The deposition velocities used in SPACCIM were taken from Ganzeveld et al. (1998), and presented in Table S2.

(Page 6, Line 164)

Reference:

Ganzeveld, L. N., Roelofs, G. J., Lelieveld, J.: A dry deposition parameterization for sulfur oxides in a chemistry and general circulation model. Journal of Geophysical Research, 103, 5679–5694, 1998.

*5. 1 – About the simulations set-up: In Table S3 and S4, the source of each value (literature or urban CAPRAM scenario) should be indicated. Why measured observations of VOC analysed from stainless steel canisters (Zhu et al., 2018) are not used to initialize the model?*

Response: We have added references for values in Table S3 and S4 as follows:

The initial chemical data include gas-phase concentrations of inorganic gases (NO, NO$_2$, O$_3$, SO$_2$, HNO$_3$, NH$_3$, H$_2$O$_2$), VOCs (including alkanes, alkenes, aromatics, aldehydes, alcohols and ketones) (Barletta et al., 2005; Duan et al., 2008; An et al., 2009; Liu et al., 2009; He et al., 2010; Ianniello et al., 2011; Meng et al., 2011; Li et al., 2011; Liu

et al., 2012b; Zhao et al., 2013b; Wang et al., 2014b; Li et al., 2015a; Li et al., 2015b; Rao, et al., 2016; Wang et al., 2016b) and particle phase data (Hu et al., 2015; Wang et al., 2015; Sun et al., 2015; Liu et al., 2014; Sun et al., 2013). In case of missing values, values are taken from the CAPRAM urban scenario (http://projects.tropos.de/capram, Herrmann et al., 2005).

(Page 6, Line 170-176)

Mt. Tai (1534 m a.s.l.), the highest mountain in the North China Plain (NCP), is influenced by air pollutants transported from the surrounding or formed along the trajectories. The measured data at Mt. Tai represents the results of transport and prior chemical processing. The aim of the present study was to investigate the processing along trajectories approaching Mt. Tai. Therefore, we have not used the measured data as input, but used VOCs data from the trajectory origin areas instead.

*6. 1 – About the simulations set-up: I do not understand aerosol parameters in Table S4. Although, the unit of ng/m³ or µ m/m³ is used in the article, aerosol species composition is given in mixing ratio (g/g) in the Table. No initial value is given for aerosol number concentrations.*

Response: We have added details about the aerosol number concentration in Table S4 (see below).

   Now the Table includes the initial mass fractions of the different aerosol components as well as the initial monodisperse physical aerosol data. Both aerosol mass and single mass fractions are used in the model to calculate the initial concentrations of the aerosol composition. The ratio g/g describes the ratio of component to the overall aerosol weight and is the standard SPACCIM input.

Table S4. Aerosol compositions and parameters applied in the SPACCIM.

| Compound | Data / $g_{compound} \, g_{aerosol}^{-1}$ | Parameter | Data |
|---|---|---|---|
| Sulfate | 0.25 | Aerosol radius | 2.0E-07 m |
| Nitrate | 0.21 | Aerosol Density | 1770 kg m$^{-3}$ |
| Ammonium | 0.16 | Aerosol number concentration | 5.1E+08 m$^{-3}$ |
| Water-soluble    organic | 0.07 | | |

| | |
|---|---|
| carbon | |
| HULIS | 0.07 |
| Water-insoluble organic carbon | 0.05 |
| Positive monovalent ions | 0.03 |
| Positive divalent ions | 0.01 |
| Metals | 0.03 |
| Elemental carbon | 0.03 |

*7. 2 – About the results: In my opinion, some important information is missing. No results are reported for the microphysical structure of simulated clouds and how they compare to observations.*

Response: The microphysical conditions of the clouds are calculated by a microphysical model using temperature, supersaturation and aerosol distribution. Alongside, we have no direct measurements of aerosol/cloud size distribution and liquid water content. Furthermore, information of the cloud measurements at Mt. Tai is not characteristic for the clouds along the trajectory. Hence, a comparison is not possible. As our study focuses on the formation of DCRCs, these are not mandatory and thus, not discussed because it is beyond the scope of the current study.

*8. Specific Revisions: Introduction: The recent following review paper should be cited: Shrivastava, M., Cappa, C. D., Fan, J., Goldstein, A. H., Guenther, A. B., Jimenez, J. L., Kuang, C., Laskin, A., Martin, S. T., Ng, N. L., Petaja, T., Pierce, J. R., Rasch, P. J., Roldin, P., Seinfeld, J. H., Shilling, J., Smith, J. N., Thornton, J. A., Volkamer, R., Wang, J., Worsnop, D. R., Zaveri, R. A., Zelenyuk, A. and Zhang, Q.: Recent advances in understanding secondary organic aerosol: Implications for global climate forcing, Rev. Geophys., 2016RG000540, doi:10.1002/2016RG000540, 2017.*

Response: We have added the paper in the introduction as follows:

SOA is also a key component of $PM_{2.5}$ and linked to adverse health effects, visibility reduction and climate change (Tabazadeh, 2005; Seagrave et al., 2006; De Gouw and Jimenez, 2009; Shrivastava et al., 2017).

(Page 2, Line 49-50)

*9. Specific Revisions: Introduction: The citation of following paper should be added in the paragraph lines 52 to 64: Mouchel-Vallon, C., Deguillaume, L., Monod, A., Perroux, H., Rose, C., Ghigo, G., Long, Y., Leriche, M., Aumont, B., Patryl, L., Armand, P. and Chaumerliac, N.: CLEPS 1.0: A new protocol for cloud aqueous phase oxidation of VOC mechanisms, Geosci. Model Dev., 10(3), 1339–1362, doi:10.5194/gmd-10-1339-2017, 2017.*

Response: We have added the paper in the introduction as follows:

Additionally, model studies show growing evidence that substantial amounts of DCRCs are formed by aqueous-phase reactions within aerosol particles, clouds and fog droplets (Sorooshian et al., 2006; Carlton et al., 2007, 2009; Ervens et al., 2008, 2011; Ervens, 2015; Tilgner and Herrmann, 2010; Tilgner et al., 2013; Mouchel-Vallon et al., 2017).

(Page 2, Line 60-62)

*10. Part 2: The sampling period of observations should be given here. A discussion about the known sources of DCRCs deduced from Zhu et al. (2018) should be done.*

Response: We have added sampling period in the revised manuscript as follows:

The sampling period was from June 4 to July 4 2014.

(Page 3, Line 86)

We have added a brief discussion about DCRC sources in the revised manuscript as follows:

Source identification indicated that DCRCs were mainly derived from anthropogenic activities followed by photochemical aging. Secondary sources, fuel combustion, photooxidation of unsaturated fatty acids and waste burning were also significant sources.

(Page 3, Line 89-91)

*11. Part 2, 2.1: The limitation of SPACCIM due to the assumptions of ideal solutions concerns deliquesced particles. This should be specified.*

Response: We have pointed out when studying deliquesced particles, the limitation of SPACCIM should be kept in mind as follows:

These limitations have to be kept in mind when studying deliquesced particles and comparing predicted and observed concentrations at Mt. Tai. The potential limitations of an ideal solution assumption compared to a non-ideal treatment are discussed in a recent paper by Rusumdar et al. (2020).

(Page 4, Line 113-115)

12. *Part 2, 2.1: Microphysical processes include in SPACCIM should be listed. Also it should be mentioned that cloud particles size distribution is spectral.*

Response: We have added a decription about the microphysical processes and cloud particles size distribution considered in the present study in Sect. 2.1 as follows:

The microphysical model applied in SPACCIM is based on the work of Simmel and Wurzler, (2006) and Simmel et al. (2005). Droplet formation, evolution and evaporation are realized by a one-dimensional sectional microphysics considering deliquesced particles and cloud droplets. In the present study, the moving bin version of SPACCIM has been applied. In the model, the growth and shrinking of aerosol particles by water vapor diffusion as well as nucleation and growth/evaporation of cloud droplets is considered. The dynamic growth rate in the condensation/evaporation process and the droplet activation is based on the Köhler theory. Due to the emphasis on complex multiphase chemistry, other microphysical processes such as impaction of aerosol particles and collision/coalescence of droplets and thus precipitation were not considered in the present study. Moreover, the air parcel model SPACCIM is not able to reflect the complexity of tropospheric mixing processes. Nevertheless, the complex model enables detailed investigations of the multiphase chemical processing of gases, deliquescent particles and cloud droplets.

(Page 4, Line 101-110)

13. *Part 2, 2.1: Does SPACCIM include aerosols? If yes, it should be specified how: which processes are considered for aerosols: microphysics processes (nucleation, aggregation, sedimentation), chemical aging, nucleation and impaction scavenging by cloud particles? Also the method to represent their size distribution should be indicated.*

*Do you use thermodynamics equilibrium to partition inorganic and organic species between gas phase and particles?*

Response: In the multiphase chemistry model, the phase transfer is treated for soluble compounds according to the resistance model of Schwartz (1986). No thermodynamics equilibrium approach is used to calculate the partitioning of soluble compounds. The model uses accommodation coefficients, gas-phase diffusion coefficients, and Henry's law constants of the phase transfer of soluble compounds (Wolke et al., 2005). The phase transfer is treated in the same manner for cloud droplets and aqueous particles following the assumption of deliquesced particles. A detailed description is already given in Wolke et al. (2005) and thus not implemented in our manuscript.

Reference:

Schwartz, S. E.: Mass-transport considerations pertinent to aqueous phase reactions of gases in liquid-water clouds, in: Chemistry of multiphase atmospheric systems. Springer, 415-471, 16, 1986.

Wolke, R., Sehili, A., Simmel, M., Knoth, O., Tilgner, A., Herrmann, H.: SPACCIM: A parcel model with detailed microphysics and complex multiphase chemistry. Atmos. Environ., 39, 4375-4388, 2005.

*14. Part 2, 2.1: Could you please indicate if recent findings on isoprene and aromatics gas phase chemistry are included in MCM? As the reference papers for MCM are from 2003, it is probably not the case and could lead to other limitations in the results.*

Response: MCMv3.3.1 systematically refined and updated the chemistry of isoprene degradation to reflect recent advances in understanding (Jenkin et al., 2015). However, in SPACCIM, MCMv3.2 was used, which made a general update of isoprene degradation chemistry, including integration of revised chemistry for isoprene-derived hydroperoxides and nitrates. Therefore, the newest findings about isoprene are not included in SPACCIM. However, regarding the review of Wennberg et al. (2018) and other new laboratory studies (e.g., Berndt et al., 2019), the isoprene oxidation scheme in MCMv3.3.1 is also not representative enough. Furthermore, the new scheme is

developed to treat more clean conditions as observed at Mt. Tai. Hence, the main oxidation product MACR and MVK should not be influenced too much by the MCMv3.2 oxidation scheme.

The degradation chemistry of aromatic VOC remains an area of particular uncertainty. Aromatic scheme in MCMv3.2 continue to use these in MCMv3.3.1. The updated aromatic VOC schemes within MCMv3.3.1 were mainly for benzene, toluene, p-xylene and 1, 3, 5-trimethylbenzene and focus on the kinetic rate constants. Therefore, the newest findings about aromatic VOC are also not included, but can be stated as minor bias for the model simulations.

The considered mechanism for isoprene and aromatics indeed impact the modeled results. In the future, we will update MCM mechanism in SPACCIM to better perform more advanced model simulation.

References:

Berndt, A. J., Hwang, J., Islam, M. D., Sihn, A., Urbas, A. M., Ku, Z., Lee, S. J., Czaplewski, D. A., Dong, M., Shao, Q., Wu, S., Guo, Z., Ryu, J. E.: Poly(sulfur-random-(1,3-diisopropenylbenzene)) based mid-wavelength infrared polarizer: Optical property experimental and theoretical analysis. Polymer, 176, 118-126, 2019.

Jenkin, M. E., Young, J. C., Rickard, A. R.: The MCM v3.3.1 degradation scheme for isoprene. Atmospheric Chemistry and Physics, 15, 11433–11459, 2015.

Wennberg, P. O., Bates, K. H., Crounse, J. D., Dodson, L. G., McVay, R. C., Mertens, L. A., Nguyen, T. B., Praske, E., Schwantes, R. H., Smarte, M. D., St Clair, J. M., Teng, A. P., Zhang, X., Seinfeld, J. H.: Gas-Phase Reactions of Isoprene and Its Major Oxidation Products. Chemical Reviews, 118, 3337-3390, 2018.

*15. Part 2, 2.2: Could you please indicate the reason to exclude days of the campaign influenced by biomass burning?*

Response: We have added reasons for excluding days of the campaign influenced by biomass burning as follows:

The aim of the study was to investigate the secondary formation of aerosol constituents

along the trajectories to Mt. Tai. However, biomass burning can be an important primary source of compounds that are often of secondary origin. Therefore, in this study, we focused on the period that was less impacted by biomass burning.

(Page 5, Line 129-133)

*16. Part 2, 2.3: You should use the proper ECCAD address; https://eccad.aeris-data.fr/. Note that you have to acknowledge ECCAD and to reference the dataset citation (see metadata page of the inventory that you used). Why did you use urban CAPRAM scenario for missing values? I suggest to indicate also initial gas-phase conditions in mixing ratio (ppbv or pptv), which is the usual quantity used for representing observations of trace gases.*

Response: (1) We have changed the ECCAD address in the revised manuscript as follows:

Biogenic emission data (isoprene, α- and β-pinenes) are obtained from Emissions of atmospheric Compounds and Compilation of Ancillary Data (ECCAD), MEGAN-MACC dataset (https://eccad.aeris-data.fr/).

(Page 5, Line 158-160)

(2) We have acknowledged ECCAD and corresponding dataset in the Acknowledgements as follows:

The authors acknowledge the Emissions of atmospheric Compounds and Compilation of Ancillary Data (ECCAD), MEGAN-MACC dataset.

(Page 21, Line 642-643)

(3) During the sampling period, $PM_{2.5}$ concentration was $98.2 \pm 29.2$ μg m$^{-3}$ (range from 37.0 to 193 μg m$^{-3}$) in daytime, and $98.6 \pm 25.3$ μg m$^{-3}$ (range from 55.7 to 143 μg m$^{-3}$) in nighttime. The $PM_{2.5}$ concentration is equal or higher than that in urban sites in European (Dimitriou and Kassomenos, 2014; Eeftens et al., 2012). Therefore, we choose urban CAPRAM scenario for missing values.

References:

Dimitriou, K., and Kassomenos, P.: A study on the reconstitution of daily PM10 and

PM2.5 levels in Paris with a multivariate linear regression model. Atmospheric Environment, 98, 648-654, 2014.

Eeftens, M., Tsai, M. Y., Ampe, C., Anwander, B., Beelen, R., Bellander, T., et al.: Spatial variation of PM2.5, PM10, PM2.5 absorbance and PM coarse concentrations between and within 20 European study areas and the relationship with NO2 - Results of the ESCAPE project. Atmospheric Environment, 62, 303-317, 2012.

(4) We have changed the unit of initial gas-phase concentrations to mixing ratio as follows:

Table S3. Initial gas-phase concentrations applied in the SPACCIM.

| Compound | Concentration | Compound | Concentration |
|---|---|---|---|
| Nitric oxide | 0.32 ppbv | p-Xylene | 94.53 pptv |
| Nitrogen dioxide | 1.72 ppbv | m-Xylene | 94.53 pptv |
| Ozone | 100.33 ppbv | Acetaldehyde | 1.00 ppbv |
| Nitric acid | 0.67 ppbv | Propionaldehyde | 70.48 pptv |
| Hydrogen peroxide | 0.31 ppbv | Butyraldehyde | 35.32 pptv |
| Formaldehyde | 0.70 ppbv | Acetone | 1.07 ppbv |
| Hydrogen | 0.46 ppmv | Methyl ethyl ketone | 29.44 pptv |
| Carbon monoxide | 1.18 ppmv | Methyl isobutyl ketone | 13.02 pptv |
| Methane | 2.06 ppmv | Glyoxal | 0.21 ppbv |
| Carbon dioxide | 332.10 ppmv | Glycolaldehyde | 0.21 ppbv |
| Sulfur dioxide | 2.14 ppbv | Methylglyoxal | 18.57 pptv |
| Ethane | 0.43 ppbv | Peroxyacetyl nitrate | 92.87 pptv |
| Propane | 80.43 pptv | Methyl hydrogen peroxide | 0.19 ppbv |
| Isoprene | 96.19 pptv | Ethyl hydrogen peroxide | 18.57 pptv |
| n-propanol | 1.30 pptv | Peroxyacetic acid | 0.19 pptv |
| Isopropanol | 51.00 pptv | Ammonia | 4.39 ppbv |
| Butanol | 0.75 pptv | Methanol | 0.42 ppbv |
| Isobutanol | 0.56 pptv | Ethanol | 0.40 ppbv |
| Ethylene glycol | 1.17 pptv | Glyoxylic acid | 0.11 ppbv |
| Ethylene | 0.96 ppbv | Glycolic acid | 0.11 ppbv |
| Toluene | 0.31 ppbv | | |
| Cresol | 0.19 pptv | | |
| o-Xylene | 62.61 pptv | | |

17. Part 3: 3.1.1: Page 6, lines 178-179, it is stated: "The reduction of OH radical is mainly caused by the reduction of the gas-phase formation pathway of the $HO_2$+ NO reaction" and page 7, lines 198-200: "Under daytime cloud droplet conditions, OH

*aqueous-phase concentrations are increased by a factor of 3, mainly due to the increased direct transfer of OH from the gas phase.". I found these statements contradictory. I guess that a plot showing the total OH concentrations (gaseous + aqueous) would clarify this point.*

Response: From the below Figure R1, we can see "The reduction of OH radical in day cloud is mainly caused by the reduction of the gas-phase formation pathway of the $HO_2$ + NO reaction". However, we forgot to mention that this sentence is related to gas-phase OH concentrations. It is changed now in the revised manuscript as follows:

The reduction of gas-phase OH radical concentrations in daytime cloud is mainly caused by the reduction of the gas-phase formation pathway of the $HO_2$+ NO reaction. (Page 7, Line 207-208)

Figure R1 also shows that "under daytime cloud droplet conditions, OH aqueous-phase concentrations are increased, mainly due to the increased direct transfer of OH from the gas phase."

Figure R2 shows time series of total OH concentrations (gaseous + aqueous phase) in the C2w case, which also suggests the two sentences are not contradicting in their revised form, now.

[Figure]

Figure R1. Modeled multiphase source and sink fluxes of gas-phase OH (above) and aqueous-phase aOH (below) oxidant (light blue column: cloud; shadow: night).

[Figure]

Figure R2. Time series of the modeled gaseous + aqueous phase OH oxidant concentrations in the C2w case (light blue column: cloud; shadow: night).

*18. Part 3: 3.1.1: Whereas, the text page 6 line 187 indicates that obtained OH and HO₂ gaseous concentrations are compared to available measurements, it is mainly modelling studies that are used for this comparison. Moreover, no details are given on these studies: which model, which conditions (period of simulation, chemical mechanism used for instance). The only observations cited show discrepancies with results, in particular for OH.*

Response: We have added period of simulation and chemical mechanism for available measurements as follows:

However, the simulated maxima of the gas-phase concentrations of OH (C2w: $3.2 \times 10^6$ molecules cm$^{-3}$, C2wo: $3.5 \times 10^6$ molecules cm$^{-3}$) and HO$_2$ (C2w: $2.9 \times 10^8$ molecules cm$^{-3}$, C2wo: $3.8 \times 10^8$ molecules cm$^{-3}$) for Mt. Tai in this study are comparable to the available measurements listed below. Compared with the modeled maximum OH ($6.0 \times 10^6$ molecules cm$^{-3}$) and HO$_2$ ($7.0 \times 10^8$ molecules cm$^{-3}$) concentrations at Mt. Tai in June 2006 using a photochemical box model- that was based on the Regional Atmospheric Chemistry Mechanism (RACM) (Kanaya et al., 2009), the OH and HO$_2$ concentrations reported here are lower. Moreover, the modeled OH and HO$_2$ concentrations in this study are lower than those of simulated results over

the Chinese megacity Beijing in August 2007 using a 1-D photochemical model (Regional chEmical and trAnsport Model, REAM-1D), whose chemistry was driven by the standard GEOS-Chem gas-phase chemistry mechanism (OH: $9 \times 10^6$ molecules cm$^{-3}$, HO$_2$: $6.8 \times 10^8$ molecules cm$^{-3}$) (Liu et al., 2012) and much lower than the measured data by laser induced fluorescence (LIF) at a rural site downwind of the megacity Guangzhou, China in 3–30 July 2006 (OH: 15-26 $\times 10^6$ molecules cm$^{-3}$, HO$_2$: 3-25 $\times 10^8$ molecules cm$^{-3}$) (Lu et al., 2012). Additionally, the simulated NO$_3$ radical maxima (C2w: $1.0 \times 10^8$ molecules cm$^{-3}$, C2wo: $1.5 \times 10^8$ molecules cm$^{-3}$) are much lower than those observed at the urban site of Shanghai, China by Differential Optical Absorption Spectroscopy (DOAS) from August 15 to October 7, 2011 ($2.5 \times 10^9$ molecules cm$^{-3}$) (Wang et al., 2013a).

(Page 7, Line 214-Page 8, Line 228)

*19. Part 3: 3.1.1: Whereas some estimations exist of OH concentration in cloud droplets (see Arakaki, T., Anastasio, C., Kuroki, Y., Nakajima, H., Okada, K., Kotani, Y., Handa, D., Azechi, S., Kimura, T., Tsuhako, A. and Miyagi, Y.: A General Scavenging Rate Constant for Reaction of Hydroxyl Radical with Organic Carbon in Atmospheric Waters, Environ. Sci. Technol., 47(15), 8196–8203, doi:10.1021/es401927b, 2013), no comparison is discussed. It seems to me that the simulated OH concentrations in droplets are high in comparison to available estimations.*

Response: We have added comparison about OH concentration in cloud droplets in the revised manuscript as follows:

Compared with OH concentrations measured in remote clouds from laboratory studies (average: $7.2 \times 10^{-15}$ mol l$^{-1}$; Arakaki et al., 2013), the modeled average aqueous-phase OH concentration in daytime clouds ($9.6 \times 10^{-14}$ mol l$^{-1}$) is much higher. The difference between measured and modeled OH concentrations is comprehensively discussed in Tilgner and Herrmann, (2018). The chapter outlined that both model results and laboratory investigations of field samples are biased. However, it should be mentioned that more comprehensive aqueous phase mechanism tends to lower OH predictions due

to higher number of possible OH sinks. On the other hand, laboratory investigations of field samples most likely tend to underestimate the OH sources due to the limitation of present offline methods. For instance, during the time period on the way from the measurement site to the laboratory, the OH radical can still be consumed by oxidation processes that cannot be resolved by the laboratory protocol, and OH sources related to the uptake of OH precursors ($H_2O_2$, ROOHs, etc.) are also excluded. Therefore, an adequate comparison is rather difficult at present.

(Page 8, Line 233- 242)

*20. Part 3: 3.1.2: I suggest using mixing ratio for gas-phase instead of concentrations when comparing simulated results with in-situ measurements. As mixing ratio is a relative quantity, it is not dependent on altitude (via pressure and temperature).*

Response: We have changed the unit of gas-phase $O_3$ and $H_2O_2$ to ppbv as follows:

In the C2wo case, measured gas-phase $O_3$ concentrations at Mt. Tai ranged from 78.6 to 108.3 ppbv (Fig. 2), which is typical in a Chinese suburban regime (Wang et al., 2013b). However, these mixing ratios are reached even at the high altitude of Mt. Tai. Additionally, the simulated maxima gas-phase $H_2O_2$ concentrations (C2w: 1.0 ppbv, C2wo: 2.3 ppbv) are lower than those observed at a rural site downwind of the more polluted area of Hebei, China (11.3 ppbv) (Wang et al., 2016b). The simulated $O_3$ maxima (C2w: 94.2 ppbv, C2wo: 105.1 ppbv) are lower than those observed at the Nanjing urban area in China (133.9 ppbv) (An et al., 2015).

(Page 9, Line 264-270)

*21. Part 3: 3.2: As no information is given on how the interactions between aerosols and cloud water are considered, it is difficult to interpret the results. Why on Fig. 3 sulphate remains constant when cloud dissipates whereas nitrate decreases at the same time?*

Response: Sulfate is a very strong acid and is the main driver for cloud droplet and aerosol acidity. When the cloud dissipates, the liquid water content is reduced by three orders of magnitude. This will decrease the pH down by a factor of three. As nitrate is

a less strong acid compared with sulfate, a higher amount of nitrate is present as nitrous acid and driven out into the gas phase, which is a common known phenomenon.

*22. Part 3: 3.3.1: I suggest recalling in legend of Fig.5 what Gly, wC2, C2, Pyr, MGly and C3 means. Does this figure show the aerosol mass concentrations as shown in Figure 3? If yes, I do not understand why the text page 11, line 316 refers to aqueous phase concentrations. Does this mean that aqueous phase concentration is the same than aerosol mass concentration? These results are difficult to interpret without knowing if thermodynamics equilibrium between gas phase and aerosols is considered for organics species. For instance, for malonic acid, it should be mainly in the aerosols and not in the gas phase (Limbeck, A., Kraxner, Y. and Puxbaum, H.: Gas to particle distribution of low molecular weight dicarboxylic acids at two different sites in central Europe (Austria), Journal of Aerosol Science, 36, 991-1005, 2005).*

Response: Figure 5 shows the aerosol mass concentrations of Gly, $\omega C_2$, $C_2$, MGly, Pyr and $C_3$. We have changed the legend of Figure 5 as follows:

Figure 5. Time series of the modeled aerosol mass concentrations of selected DCRCs (top: Gly, $\omega C_2$, $C_2$; bottom: MGly, Pyr, $C_3$) in the C2w and C2wo cases (light blue column: cloud; shadow: night; green triangle: the maximum (above), average (middle) and minimum (below) values of measured concentrations at Mt. Tai).

(Page 39)

We have changed "aqueous-phase" in page 11, line 316 to "aerosol mass" in the revised manuscript as follows:

Figure 5 shows the modeled aerosol mass concentrations of Gly, $\omega C_2$, $C_2$, methylglyoxal (MGly), Pyr, and malonic acid ($C_3$) both in the C2w and C2wo cases as well as the values measured at Mt. Tai.

(Page 12, Line 361-362)

*23. Part 3: 3.3.1: Dicarbonyl compounds: I find that the remarkable result is that Gly and MGly are very similar concentrations at the end of the simulation considering or not cloud chemistry. In addition, the small increase of MGly in the cloud case in*

*comparison to the no cloud one, seems to be related to its small production inside cloud droplets during night. So, I disagree with this statement: "This might have been caused by the fact that the aqueous oxidation fluxes under nighttime cloud conditions are lower than the ones under daytime".*

Response: We have added sentence about similar concentrations of Gly or MGly at the end of the simulation with or without cloud as follows:

It's worth noting that Gly or MGly have similar concentrations at the end of the simulation with or without cloud chemistry.

(Page 12, Line 374-375)

However, OH radical concentrations in the nighttime cloud are one order of magnitude lower than those during daytime cloud (see Figure R2). The result leads a much lower oxidation rate of the OH radical under nighttime cloud conditions (see Figure R1). Furthermore, the increased aqueous-phase $NO_3$ radical concentrations are not comparable with the decreased aqueous-phase OH radical concentrations (see Figure S3 in the supplement). Thus, slightly increased MGly is related to the lower nighttime MGly oxidation rate. We have changed corresponding sentence in the revised manuscript as follows:

This might have been caused by the fact that the aqueous oxidation fluxes under nighttime cloud conditions are lower than the ones under daytime, because of much lower OH radical concentrations under nighttime cloud conditions (Fig. S3).

(Page 12, Line 370-372)

*24. Part 3: 3.3.1: C2 carboxylic acids: Could you please rewrite this passage, which is not clear, only describes the curves and does not give hypotheses about the observed trends.*

Response: We have added sentence in the revised manuscript as follows:

Compared with the C2wo case, the C2w case shows higher $\omega C_2$ concentrations, which suggests that cloud processes play important roles in $\omega C_2$ formation and oxidation.

(Page 12, Line 380-381)

*25. Part 3: 3.3.1: C3 carboxylic acids: I don't see an increase of Pyr in the nighttime cloud.*

Response: During the nighttime cloud, although having a sharp increase and decrease at the beginning, Pyr concentrations are indeed increased. For example, during nighttime cloud time interval ranging from 24-24.96 h, Pyr concentrations are increased from 28.3 to 33.6 ng m$^{-3}$; during nighttime cloud time interval ranging from 47.3-48.96 h, Pyr concentrations are increased from 122.2 to 125.6 ng m$^{-3}$; during nighttime cloud time interval ranging from 71.3-72.96 h, Pyr concentrations are increased from 182.5 to 185.0 ng m$^{-3}$.

*26. Part 3: 3.3.2: As described in Zhu et al. (2018), DCRCs observations are available for daytime and nighttime. Thus, I suggest indicating in Table 3, ratio for day and night. Could you please specify on which simulated period the averages are done (the three days or only the third day)?*

Response: We only use the third day data to calculate the ratio of the modeled and measured DCRC compounds, and Table 3 has been changed as follows:

Table 3. Ratios of the concentrations of the modeled and measured DCRC compounds in the different model trajectories at Mt. Tai.

| Compound | Model case | | | |
| --- | --- | --- | --- | --- |
| | C2w-day | C2w-night | C2wo-day | C2wo-night |
| $C_2$ | 0.30 | 0.27 | 0.23 | 0.21 |
| $\omega C_2$ | 7.07 | 6.94 | 3.35 | 3.43 |
| $C_3$ | 1.82 | 1.86 | 1.57 | 1.58 |
| Pyr | 8.95 | 7.12 | 4.34 | 3.22 |
| Gly | 0.19 | 0.23 | 0.13 | 0.16 |
| MGly | 2.30E-3 | 2.72E-3 | 1.35E-3 | 1.67E-3 |

(Page 33-34)

*27. Part 3: 3.3.2: By this sentence "SPACCIM overestimates the measured ωC2 concentrations, but underestimates the measured C2 ones, suggesting the conversion of ωC2 might be implemented less efficiently into CAPRAM.", do you mean that the aqueous phase oxidation of ωC2 producing C2 implemented in CAPRAM seems to be less efficient than in the field?*

Response: Yes. We have rephrased the sentence to make it clear as follows:

SPACCIM overestimates the measured $\omega C_2$ concentrations, but underestimates the measured $C_2$ ones, suggesting that the conversion of $\omega C_2$ is implemented less efficiently into CAPRAM than it is seen in the field.

(Page 13, Line 404-406)

28. *Part 3: 3.3.2: Page 12, lines 356-362: the hypothesis of missing processes enhancing the partitioning of MGly implies that the model is capable of simulating realistic total MGly concentrations (gas phase + aerosols): is-it the case?*

Response: Missing partitioning processes of MGly is an important reason for the extremely low MGly concentration in model simulation, which is about three orders of magnitude lower than the measured data. Even if complementing the missing MGly partitioning processes, SPACCIM model may not simulate realistic MGly concentrations in the field observation. Because other factors, such as model input data (e.g. emission inventory, initial gas- and aqueous-phase concentrations of key species at originated areas), also impact the modeled MGly concentration. We have changed the corresponding sentence in the revised manuscript as follows:

Phase partitioning between gas and aqueous phase, a key process for modeled MGly concentration, may be not sufficient enough to predict the measured MGly aerosol concentrations in the field because of model simplicity.

(Page 13, Line 406-408)

29. *Part 3: 3.3.2: I disagree with this statement "The emission data are obtained through model calculations, not field measurements". Emission data comes from inventory, which is developed in part based upon measurements. I guess that the more probable error coming from emission data is due to the horizontal and temporal resolution of inventories used. This point should be discussed.*

Response: We have changed "model calculations" to "emission inventory", and added the impact of horizontal and temporal resolution of inventory in the revised manuscript as follows:

The emission data are obtained through a new anthropogenic emission inventory in Asia, which provides monthly emissions in 2010 by sector at 0.25° × 0.25° resolution. However, the model simulation period in this study is in 2014 and the spatial resolution is less than 5° × 2°. Therefore, errors in conversion of the emission data can occur. (Page 13, Line 416-419)

*30. Part 3: 3.3.2: Why do you mean by "The height of Mt. Tai (about 1500 m) also causes its input to be indefinite."?*

Response: Anthropogenic sources in emission inventory, such as power, industry, residential and agriculture, emitted near the Earth's surface. However, they can be transported over long distances and a wider area. A key factor on the transport is the boundary layer, which is the lowest atmospheric layer immediately affected by the Earth's surface. Anthropogenic pollution generates a strong lid on the top of the boundary layer, hindering turbulent mixing of pollutants from the surface to higher up. Aerosol particles increase the boundary layer stability and cause any subsequent emissions to remain in a lower boundary layer, reduce the mixing height even further. The boundary layer height varies from a few dozen meters to a few kilometers. Therefore, the height of Mt. Tai (about 1500 m) also causes its input to be indefinite.

*31. Part 3: 3.3.2: MCM mechanism: I guess you could cite other limitations, in particular in biogenic VOC oxidation as the Mt. Tai is surrounded by deciduous forest.*

Response: This is correct, but the height of Mt. Tai is more than 1500 m, and often above the planetary boundary layer. Therefore, it is decoupled from direct ground emissions most of the time (Zhu et al. 2018). Hence, these missing values are low in their limitation and not discussed further.

Reference:

Zhu, Y., Yang, L., Chen, J., Kawamura, K., Sato, M., Tilgner, A., van Pinxteren, D., Chen, Y., Xue, L., Wang, X., Simpson I. J., Herrmann, H., Blake D. R., and Wang, W. X.: Molecular distributions of dicarboxylic acids, oxocarboxylic acids and α-

dicarbonyls in PM2.5 collected at the top of Mt. Tai, North China, during the wheat burning season of 2014, Atmos. Chem. Phys., 18, 10741-10758, 2018.

*32. Part 3: 3.3.2: I disagree that ratios are acceptable except for C3. I think that it is an interesting result to show that, even a sophisticated model as the one used in this study, is not able to reproduce DCRCs observations. A discussion trying to assess why the C3 ratio is close to 1 should be interesting.*

Response: We have deleted "which can be regarded as an acceptable range due to the model and input data limitations" in the manuscript.

(Page 14, Line 436-437)

A discussion about the reasons of $C_3$ ratio is close to 1 has been added in the revised manuscript as follows:

Interestingly, the ratio of $C_3$ is close to 1, which might be related to a good representation of the concentrations of $C_3$ precursors. The comparison indicates that formation pathways of DCRCs implemented in CAPRAM4.0 is realistic, but highly dependent on the input data of precursors.

(Page 14, Line 437-439)

*33. Part 3: 3.4.1: $\omega C2$: I see a low net formation flux during the last night for non-cloud period.*

Response: We have checked the net formation flux and net sink flux of $\omega C_2$, and found really an extremely low net formation flux during the non-cloud period. We have changed the corresponding sentences in the revised manuscript as follows:

The results reveal a net formation flux that mainly occurs during cloud conditions as well as a net degradation mainly during non-cloud periods. About 94 % of the net formation flux of $\omega C_2$ is simulated under cloud condition. However, the non-cloud conditions represent 99 % of the net sink flux of $\omega C_2$.

(Page 15, Line 464-467)

*34. Part 3: 3.4.2: C2: Please moderate the last sentence: one of the reasons and not the reason.*

Response: We have changed "the reason" to "one of the reasons" in the revised manuscript as follows:

The possible overestimation of the photolytic decay leads to a significantly low $C_2$ concentration and might be thus one of the reasons for the underestimated $C_2$ concentration.

(Page 16, Line 501-503)

*35. Part 3: 3.4.3: Pyr: Do you consider the photolysis of Pyr in aqueous phase (Reed Harris, A. E., Ervens, B., Shoemaker, R. K., Kroll, J. A., Rapf, R. J., Griffith, E. C., Monod, A. and Vaida, V.: Photochemical Kinetics of Pyruvic Acid in Aqueous Solution, J. Phys. Chem. A, doi:10.1021/jp502186q, 2014.) ?*

Response: No, we don't consider the Pyr photolysis in the present aqueous phase mechanism. It will be considered in CAPRAM in the future. However, it needs to be mentioned that a large fraction of Pyr is in the hydrated form in acidic solutions. Therefore, the photochemical active carbonyl group is deactivated and other processes might play more important roles in aqueous solution for the degradation of Pyr, such as $H_2O_2$ reaction.

*36. Part 3: 3.5.1: Green lines on Fig. 7 are difficult to see: could you please use another colour (grey for instance)?*

Response: We have changed green lines to tawny lines in Figure 7 as follows:

[Figure]

Figure 7. Concentration variations of modeled sulfate, nitrate, ammonium, Gly, $\omega C_2$, $C_2$, MGly, Pyr and $C_3$ when doubling emission data (light blue column: cloud; shadow: night).

(Page 41-42)

*37. Part 3: 3.5.2: Could you please explain how negative and positive RIR values have to be interpreted? It would help the reader to follow the discussion on DCRCs precursors. Page 17, line 519: please suppress As.*

Response: We have explained positive and negative RIR values in the revised manuscript as follows:

The positive or negative RIR value reveals that reducing precursor emissions would weaken or aggravate DCRCs formation, respectively.

(Page 18, Line 554-555)

We have deleted "As" as follows:

For Pyr, in both C2w and C2wo cases, alkenes are the dominant precursor group with the largest RIRs.

(Page 18, Line 575)

---

## Author Comment (AC3)

[revised manuscript text omitted]

1105 **Figure 4. Modeled multiphase (gas + aqueous phase) source and sink fluxes of sulfate and nitrate (light blue column: cloud; shadow: night; percent is for the third model day).**

[Figure]

**Figure 5. Time series of the modeled aerosol mass concentrations of selected DCRCs (top: Gly, ωC$_2$, C$_2$; bottom: MGly, Pyr and C$_3$) in the C2w and C2wo cases (light blue column: cloud; shadow: night; green triangle: the maximum (above), average (middle) and minimum (below) value of measured concentrations at Mt. Tai).**

[Figure]

**Figure 6. Modeled multiphase (gas + aqueous phase) source and sink fluxes of ωC₂ (above left), C₂ (above right), Pyr (below left) and C₃ (below right) along the trajectory of the third model day (light blue column: cloud; shadow: night; percentage is for the third model day).**

[Figure]

[Figure]

... [2]

**Figure 7. Concentration variations of modeled sulfate, nitrate, ammonium, Gly, ωC₂, C₂, MGly, Pyr and C₃ when doubling emission data (light blue column: cloud; shadow: night).**

[Figure]

**Figure 8. The calculated RIRs for C$_2$, Pyr and ωC$_2$ in both the C2w (green bars) and C2wo (blue bars) cases at Mt. Tai (column: RIR values; red dots: emission data).**

[Figure]

**Figure 9. Correlations between the decreasing ratios of radical oxidants and C₂-RIR (above) and ωC₂-RIR (below) under C2w and C2wo conditions, respectively.**

[Figure]

**Figure 10. Concentration variations of modeled Gly, ωC₂ and C₂ when increasing Gly' Henry law constant by two orders of magnitude (shadow: night; K_H: Henry law constant).**

1155

| Seite 32: [1] Gelöscht | DELL | 28.02.20 11:22:00 |
|---|---|---|
| Seite 40: [2] Gelöscht | DELL | 28.02.20 11:28:00 |

---

## Referee Report (RR1)

**Review for revised version of:**
*"Multiphase MCM/CAPRAM modeling of formation and processing of secondary aerosol constituents observed at the Mt. Tai summer campaign 2014"*
*by*
*Y. Zhu et al.*

*Submitted to Atmospheric Chemistry and Physics*

**General Comments:**
I thank the authors for their detailed answers about my concerns. However, I have still some remaining concerns on the revised manuscript detailed below.

**Overall recommendation:**
I recommend that the paper should be accepted for publication in Atmospheric Chemistry and Physics after major and specific revisions listed below.

**Major Comments:**
I disagree with this statement in author's response: "MEGAN-MACC includes21 biogenic species, such as ethane, propane, propene, toluene. However, most of the treated compounds are mainly from anthropogenic sources, especially in China." For instance, emissions of methanol (acetone) from biogenic sources are of the same order of magnitude than emission of total alcohols (ketones) from anthropogenic sources in the vicinity of Mt Tai for June 2014. This can be verified using the ECCAD database comparing inventories CAMS-GLOB-BIO and CAMS-GLOB-ANT.

**Specific Revisions:**
**Part 2.1**: About my comment 13: "Does SPACCIM include aerosols? If yes, it should be specified how: which processes are considered for aerosols: microphysics processes (nucleation, aggregation, sedimentation), chemical aging, nucleation and impaction scavenging by cloud particles? Also the method to represent their size distribution should be indicated. Do you use thermodynamics equilibrium to partition inorganic and organic species between gas phase and particles?" The author did not answer to this point. I knew that exchange of soluble gases between gas phase and liquid phase is considered following Schwartz (1986) in SPACCIM. My point is about the treatment of physical and thermodynamical processes concerning aerosols.

**Part 3.1.1**: About my comment 18: "Whereas, the text page 6 line 187 indicates that obtained OH and HO2 gaseous concentrations are compared to available measurements, it is mainly modelling studies that are used for this comparison. Moreover, no details are given on these studies: which model, which conditions (period of simulation, chemical mechanism used for instance). The only observations cited show discrepancies with results, in particular for OH.", now simulated studies are well detailed but, in my opinion, the sentence (page 7, lines 240-243) should be rewritten. Indeed, I didn't find that maxima of simulated results are comparable with available measurements, especially for OH. Moreover, you should explain that these results are discussed in comparison to measurements, but also to previous modelling studies.

---

## Author Response (AR2)

**Leibniz Institute for Tropospheric Research**

**Prof. Dr. Hartmut Herrmann**
Head of TROPOS Atmospheric
Chemistry Department (ACD)
herrmann@tropos.de
phone: +49 341 2717 7024
fax: +49 341 2717 7012
Permoserstraße 15
04318 Leipzig (Germany)

Leibniz-Institut für Troposphärenforschung  Permoserstraße 15  D-04318 Leipzig

To:
Copernicus Gesellschaft mbH
Bahnhofsallee 1e
37081 Göttingen
Germany

03-04-2020

**Manuscript for Atmospheric Chemistry and Physics Discussions (MS No.: acp-2019-982)**
**"Multiphase MCM/CAPRAM modeling of formation and processing of secondary aerosol constituents observed at the Mt. Tai summer campaign 2014" by Zhu et al.**

Dear Prof. Dr. Holger Tost,

please find attached our answers to the reviewer comments for the manuscript mentioned above together with its revised version. The authors would like to thank reviewer for the constructive and good suggestions to further improve our manuscript. We have considered the reviewer comments and revised the manuscript. Below, we provide responses to the comments in blue, with changes made in the manuscript highlighted in red.

Yours sincerely,

Prof. Dr. Hartmut Herrmann
Head of Atmospheric Chemistry Department at Leibniz Institute for
Tropospheric Research
Leipzig, Germany

Prof. Dr. Likun Xue
Environment Research Institute
Shandong University,
Ji'nan, Shandong, China

Leibniz Institute for Tropospheric Research
Phone: +49 341 235-3210
Fax: +49 341 235-2139
info@tropos.de
http://www.tropos.de

Commerzbank Leipzig
Account No: 102 14 50
Sort Code: 860 400 00
IBAN: DE77 8604 0000 0102 1450 00
SWIFT CODE: COBADEFF 860

Mitglied der

[Figure]

Leibniz-Gemeinschaft

**Responses to the reviewer comments on**

**"Multiphase MCM/CAPRAM modeling of formation and processing of secondary aerosol constituents observed at the Mt. Tai summer campaign 2014" by Zhu et al.**

The authors would like to thank reviewer for the good suggestions obtained in the review and for giving us the chance to further improve our manuscript! We have carefully considered all of your comments and revised the manuscript accordingly. Below, we provide the point-to-point responses to your comments in blue, with changes made in the manuscript highlighted in red.

**Response to Reviewer:**

*I thank the authors for their detailed answers about my concerns. However, I have still some remaining concerns on the revised manuscript detailed below. I recommend that the paper should be accepted for publication in Atmospheric Chemistry and Physics after major and specific revisions listed below.*

Response: We appreciate the reviewer for the helpful comments and suggestions. Below, we address the comments and have revised the manuscript accordingly. For clarity, the reviewer's comments are listed below in *black italics*, while our responses and changes in the manuscript are shown in blue and red, respectively.

*1. I disagree with this statement in author's response: "MEGAN-MACC includes 21 biogenic species, such as ethane, propane, propene, toluene. However, most of the treated compounds are mainly from anthropogenic sources, especially in China." For instance, emissions of methanol (acetone) from biogenic sources are of the same order of magnitude than emission of total alcohols (ketones) from anthropogenic sources in the vicinity of Mt Tai for June 2014. This can be verified using the ECCAD database comparing inventories CAMS-GLOB-BIO and CAMS-GLOB-ANT.*

Response: In previous response,"MEGAN-MACC includes 21 biogenic species, such as ethane, propane, propene, toluene. However, most of the treated compounds are

mainly from anthropogenic sources , especially in China." may have somehow misleading. We want to express that "MEGAN-MACC includes 21 biogenic species. However, some of these compounds, such as ethane, propane, propene, toluene, are mainly emitted from anthropogenic sources, especially in China.".

Acetone is emitted into the atmosphere from both natural and anthropogenic sources. Natural sources include direct emissions from vegetation, decaying organic material and secondary production by the oxidation of biogenic hydrocarbons (Singh et al., 1994; Jacob et al., 2002). Anthropogenic sources are vehicular emissions, solvent use and secondary production by the oxidation of man-made hydrocarbons (Singh et al., 1994; Jacob et al., 2002).

Recently, many studies about acetone sources in China have found that ambient acetone was mainly caused by vehicular emission, not biogenic sources (Guo et al., 2013; Chi et al., 2008; Huang et al., 2008).

Methanol, as the important oxygenated volatile organic compounds (OVOCs) species, also has natural and anthropogenic sources. Huang et al. (2020) found that anthropogenic sources contributed 58% to methanol, while biogenic sources contributed 24% in North and South China. Huang et al. (2019) reported that anthropogenic primary source contributed 73% for methanol.

Therefore, emission data of acetone and methanol didn't only represent biogenic sources, mainly suggest anthropogenic sources. We are aware that this can cause uncertainties. However, our study focuses on formation of important organic acids that contribute to SOA formation. Both acetone and methanol are not known to contribute significantly to the formation of such acids (Tilgner and Herrmann, 2010). Furthermore, because of the very low reaction rate constants in comparison to VOCs and other OVOCs (around $10^{-13}$ $cm^3$ molecules$^{-1}$ s$^{-1}$ for methanol and acetone IUPAC, http://iupac.pole-ether.fr/), the effect on the $HO_x$ budget would also be minor. Hence, the bias is expected to be small.

As our previous response, propane and butane is the main components of liquefied petroleum gas/natural gas (McCarthy et al., 2013); butene is the main constituents of

gasoline (Xie and Berkowitz, 2006; Brown et al., 2007); ethane, propene and toluene originate from automobile exhaust (Chang et al., 2009); $C_3$-$C_6$ alkanes are associated with unburned vehicular emissions (Grosjean et al., 1999; Guo et al., 2004); ethene indicated its relation to coal and biomass burning (Liu et al., 2008); formaldehyde has 43% contribution from anthropogenic sources compared with 42% from biogenic sources (Huang et al., 2020); acetaldehyde has 45% contribution from anthropogenic sources compared with 37% from biogenic sources (Huang et al., 2020); formic acid and acetic acid are mostly primary and secondary products by the oxidation of various organic precursors of anthropogenic origin in North China Plain (Mochizuki et al., 2017); ketones and aldehydes are produced from anthropogenic secondary oxidation (Liu et al., 2015).

Therefore, emission data of these species are obtained from a new anthropogenic emission inventory in Asia (Li et al., 2017), not from MEGAN-MACC.

Response: The authors thank the reviewer for this comment. According to the reviewer comment we have extended the "Model and mechanism description" section. There, it is now explicitly outlined that the applied SPACCIM model includes a cloud microphysical model only that of course treats aerosol particles. However, it doesn't focus on smaller aerosol particles and aerosol microphysical processes such as nucleation, aggregation etc. This issue is now stated in the revised model description.

As shown in SI, we have performed simulations with monodispersed aerosol particles and not with a ploy-dispersed distribution. Therefore, no method to represent an aerosol size distribution is described.

As mentioned in the revised manuscript, the treatment of the phase partitioning is considered in SPACCIM following the approach by Schwartz (1986). This approach is a kinetic approach implying that the transfer fluxes are calculated at each time step and no thermodynamics equilibrium of inorganic and organic compounds between gas phase and particles/droplets is assumed in SPACCIM. For the sake of clarity, the description in the manuscript has been updated indicating the kinetic treatment of the phase transfer in SPACCIM.

SPACCIM combines a multiphase chemical model with a cloud microphysical model, simulating aqueous-phase chemistry in deliquesced particles and cloud droplets. The cloud microphysical model applied in SPACCIM is based on the work of Simmel and Wurzler, (2006) and Simmel et al. (2005).

(Page 4, Line 97-100)

However, SPACCIM cannot assess the complexity of (i) the tropospheric mixing processes along the transport, (ii) occurring aerosol particle microphysical processes (e.g., nucleation, aggregation, etc.) and (iii) the effects of non-ideal solutions on the occurring multiphase chemistry.

(Page 4, Line 109-112)

Phase exchange processes (in total 275) are implemented based on the kinetic resistance model of Schwartz (1986), in which the mass accommodation coefficient, the gas phase diffusion coefficient and the Henry's law constant are considered.

(Page 4, Line 119-121)

*3. Part 3.1.1: About my comment 18: "Whereas, the text page 6 line 187 indicates that obtained OH and HO2 gaseous concentrations are compared to available measurements, it is mainly modelling studies that are used for this comparison.*

*Moreover, no details are given on these studies: which model, which conditions (period of simulation, chemical mechanism used for instance). The only observations cited show discrepancies with results, in particular for OH.", now simulated studies are well detailed but, in my opinion, the sentence (page 7, lines 240-243) should be rewritten. Indeed, I didn't find that maxima of simulated results are comparable with available measurements, especially for OH. Moreover, you should explain that these results are discussed in comparison to measurements, but also to previous modelling studies.*

Response: We have rewritten the corresponding sentence as follows:

[revised manuscript text omitted]